# The generation of HepG2 transmitochondrial cybrids to reveal the role of mitochondrial genotype in idiosyncratic drug-induced liver injury

Amy Louise Ball[1], Carol E Jolly[1], Mark G Lennon[2], Jonathan J Lyon[3], Ana Alfirevic[4], Amy E Chadwick[1]*

[1]Centre for Drug Safety Science, Department of Pharmacology and Therapeutics, University of Liverpool, Liverpool, United Kingdom; [2]GSK, BioStatistics, Stevenage, United Kingdom; [3]GSK, Safety Assessment, Ware, United Kingdom; [4]The Wolfson Centre for Personalised Medicine, Department Pharmacology and Therapeutics, University of Liverpool, Liverpool, United Kingdom

## Abstract

**Background:** Evidence supports an important link between mitochondrial DNA (mtDNA) variation and adverse drug reactions such as idiosyncratic drug-induced liver injury (iDILI). Here, we describe the generation of HepG2-derived transmitochondrial cybrids, to investigate the impact of mtDNA variation on mitochondrial (dys)function and susceptibility to iDILI. This study created 10 cybrid cell lines, each containing distinct mitochondrial genotypes of haplogroup H or haplogroup J backgrounds.

**Methods:** HepG2 cells were depleted of mtDNA to make rho zero cells, before the introduction of known mitochondrial genotypes using platelets from healthy volunteers (n=10), thus generating 10 transmitochondrial cybrid cell lines. The mitochondrial function of each was assessed at basal state and following treatment with compounds associated with iDILI; flutamide, 2-hydroxyflutamide, and tolcapone, and their less toxic counterparts bicalutamide and entacapone utilizing ATP assays and extracellular flux analysis.

**Results:** Whilst only slight variations in basal mitochondrial function were observed between haplogroups H and J, haplogroup-specific responses were observed to the mitotoxic drugs. Haplogroup J showed increased susceptibility to inhibition by flutamide, 2-hydroxyflutamide, and tolcapone, via effects on selected mitochondrial complexes (I and II), and an uncoupling of the respiratory chain.

**Conclusions:** This study demonstrates that HepG2 transmitochondrial cybrids can be created to contain the mitochondrial genotype of any individual of interest. This provides a practical and reproducible system to investigate the cellular consequences of variation in the mitochondrial genome, against a constant nuclear background. Additionally, the results show that inter-individual variation in mitochondrial haplogroup may be a factor in determining sensitivity to mitochondrial toxicants.

**Funding:** This work was supported by the Centre for Drug Safety Science supported by the Medical Research Council, United Kingdom (Grant Number G0700654); and GlaxoSmithKline as part of an MRC-CASE studentship (grant number MR/L006758/1).

## Editor's evaluation

*For correspondence:
Amy.Chadwick@liverpool.ac.uk

The aim of this study was to demonstrate the role of varying mitochondrial DNA levels as an important factor in drug-induced cell injury, hence creating a novel in vitro model which was representative of the diversity in mitochondrial genotype. The authors take a clever approach by using cybrid cell lines to test the role of mtDNA variations, both mutational load, and DNA level, and propose that such cell models could be potentially representative of mitochondrial genome diversity. Their findings provide evidence that these mechanisms could play a role in individual susceptibility to hepatic adverse drug reactions. This also adds an important understanding of the role of mitochondria in the onset of drug-induced toxicity.

## Introduction

Drug-induced liver injury (DILI) is a leading cause of acute liver failure in the western world (*Bernal and Wendon, 2013*; *Tujios and Lee, 2018*). Although it is rare (19.1 cases per 100,000 inhabitants), it is a major cause of drug withdrawal due to its associated morbidity and mortality (*Leise et al., 2014*). DILI can be broadly divided into two categories, idiosyncratic and intrinsic injury. iDILI is characterized by a complex dose-response relationship, a lack of predictivity from the primary pharmacology of a drug, and significant inter-individual variation. This means that, despite being less common than intrinsic injury, iDILI can be argued to have far greater economic and personal costs (*Fontana, 2014*).

Drug-induced mitochondrial dysfunction is one of the mechanisms implicated in the onset of DILI. This is supported by the fact that 50% of drugs with a black box warning for hepatotoxicity also contain a mitochondrial liability, with many associated specifically with iDILI (*Dykens and Will, 2007*). It is hypothesized that mitochondria may be an important source of the inter-individual variation that underpins susceptibility to iDILI. Specifically, mitochondria contain their own genome, mtDNA, which is distinct from nuclear DNA. Single nucleotide polymorphisms (SNPs) in mtDNA are often inherited together forming haplogroups, defined as groups of people with a common mitochondrial DNA lineage that shares distinct patterns of mitochondrial SNPs. Not only have associations between specific haplogroups and mitochondrial function been determined, but haplogroups have also been associated with specific adverse drug reactions and drug efficacy (*Jones et al., 2021a*). However, in in vitro research, the presence of a variable nuclear genome makes clear relationships with mitochondrial genotype difficult to define. In previous research, we have circumvented this issue by using freshly isolated platelets as a model system, demonstrating that mitochondrial genotype-specific responses to drugs are present at the haplogroup level (*Ball et al., 2021*). However, such an approach is not practical for reproducible mechanistic toxicology or preclinical screening studies due to issues of inter-individual and day-to-day variability, as well as sample availability. Therefore, alternative hepatic models are required to probe the effect of mitochondrial genotype on the onset of iDILI.

Despite limitations regarding physiological relevance, HepG2 cells remain valuable as one of the most commonly used preclinical cell lines for the in vitro assessment of DILI (*Weaver et al., 2020*). One of the drawbacks of their use in detecting iDILI is, as a homogenous cell line they do not represent inter-individual variation, including within the mitochondrial genome. The importance of assessing the role of inter-individual variation on patient susceptibility to adverse drug reactions is well recognized, and the potential role of mtDNA variants could prove vital in drug development and safety (*Jones et al., 2021b*; *Penman et al., 2020*). However, this area remains under-researched due to previously mentioned limitations in the current in vitro models and the economic and practical limitations of recruiting patients for clinical trials. Therefore, there is a pressing need for new strategies to explore the links between mtDNA and iDILI, whilst also enabling inter-individual variation to be accounted for at the preclinical stage (*Fermini et al., 2018*; *Jones et al., 2021b*).

Here, we describe the generation and testing of transmitochondrial cybrids (herein referred to as cybrids) as a reproducible in vitro model of personalized mtDNA variation. Cybrids are typically produced by the fusion of enucleated cells (cytoplasts) or anucleate cells (e.g. platelets) with cells that have been depleted of their mtDNA (rho zero [$\rho$0] cells) (*King and Attardi, 1989*). When cybrids are generated using the same population of $\rho$0 cells, fused with anucleate cells harboring different mtDNA variants, the effects of mtDNA can be assessed against a stable nuclear background (*Wilkins et al., 2014*; *Penman et al., 2020*). Excitingly, the inclusion of mtDNA variation made possible using HepG2 cybrids may offer an enhanced preclinical prediction of iDILI, by helping to reveal the

mechanistic basis of differences associated with mtDNA variants. However, to date, there have been no reports of the generation of cybrids from a HepG2 p0 cell line (*Bale et al., 2014*).

This study aimed to generate a panel of HepG2 transmitochondrial cybrids, and utilize them in a proof of principle study, to investigate the effect of mtDNA variants upon mitochondrial function and susceptibility to drug-induced mitochondrial dysfunction. Despite the acknowledged limitations of the HepG2 model, they remain an important mainstay of pharmaceutical preclinical testing for hepatotoxicity (*Weaver et al., 2020*). Therefore, their conversion into cybrid cells is highly relevant and useful to both the pharmaceutical industry and academic research. Ten distinct populations of HepG2 cybrids were created using platelets from 10 healthy volunteers of known haplogroup; five cell lines containing each containing a different haplogroup H genome and five containing different haplogroup J genome. Haplogroup H was selected for study as it is the most common haplogroup in the UK (*Eupedia, 2016*). Haplogroup J, on the other hand, is less common in the UK but is characterized by non-synonymous mutations in regions of the mitochondrial genome that encode respiratory complex I. Although no studies have specifically reported on iDILI and mitochondria haplogroup, haplogroup J is also reported to be more susceptible to mitochondrial toxins (*Ghelli et al., 2009*; *van Oven and Kayser, 2009*; *Eupedia, 2016*; *Strobbe et al., 2018*).

The effects of mtDNA variation on drug-induced mitochondrial dysfunction were assessed by the treatment of cybrids with a panel of compounds with a known clinical association with DILI and proven mitochondrial liabilities; flutamide, 2-hydroxyflutamide, and tolcapone. The non-hepatotoxic structural counterparts bicalutamide and entacapone were also investigated for comparison. Flutamide, a non-steroidal antiandrogen for the treatment of prostate cancer, has a boxed warning for hepatotoxicity and is a known inhibitor of mitochondrial complex I (*Coe et al., 2007*). 2-hydroxyflutamide, is the primary human metabolite of flutamide and is a known inhibitor of both respiratory complexes I and II. In humans, the rapid first-pass metabolism of flutamide results in much higher maximum serum concentrations of 2-hydroxyflutamide than its parent compound (4400 nM vs 72.2 nM). However, HepG2 cells have a limited expression of many of the enzymes required for xenobiotic metabolism, including cytochrome P450 1A2, which is the primary route for the generation of 2-hydroxyflutamide. Therefore, in addition to the parent compound, the cybrids were also dosed directly with 2-hydroxyflutamide (*Shet et al., 1997*; *Sison-Young et al., 2015*; *Ball et al., 2016*). Meanwhile, tolcapone, a catechol-o-methyl transferase inhibitor used to treat Parkinson's disease, was withdrawn due to cases of liver failure and is a known uncoupler of oxidative phosphorylation (*Olanow, 2000*; *Benabou and Waters, 2003*; Olanow and Watkin.)

Generating a panel of personalized transmitochondrial cybrids offers a novel and unique in vitro model for idiosyncratic hepatotoxicity, allowing us to begin to map the functional effects of mitochondrial genotype. See *Figure 1* for a schematic overview of the study.

## Results and discussion

### At basal state, the oxidative phosphorylation activity of haplogroup H and haplogroup J cybrids differed at the level of individual complex activity, specifically complex I and II

No significant differences in oxidative phosphorylation parameters were observed between haplogroup H and J cybrids (*Figure 2A*). This may arise from the recruitment of only healthy volunteers for this study, with no clinical phenotype of mitochondrial dysfunction. However, importantly, the recruitment of such healthy volunteers is representative of the clinical situation, as individuals who experience iDILI tend not to display a pre-existing phenotype of mitochondrial dysfunction before treatment. The authors also recognize that due to this, the concentrations of drugs used were higher than the reported $C_{max}$ of the compounds (flutamide $C_{max}$ – 72.2 nM, 2-hydroxyflutamide $C_{max}$ – 4.4 µM, bicalutamide 1.7 µM, tolcapone 16.5 µM, and entacapone 3.9 µM) (*Schulz et al., 1988*; *Cockshott, 2004*; *Khetani et al., 2013*). However, these concentrations are elevated in order to try to imitate processes in more susceptible individuals.

Individual complex-driven maximal respiration, herein referred to as complex activity, was measured in permeabilized cells (*Figure 2B*). Complex I and II-driven maximal respiration were significantly greater in haplogroup J cybrids (complex I mean difference 2.46 pmol/min/ug protein, 95% CI [0.4, 4.53], p=0.021; complex II mean difference 2.97 pmol/min/ug protein, 95% CI [0.9, 5.04], p=0.006).

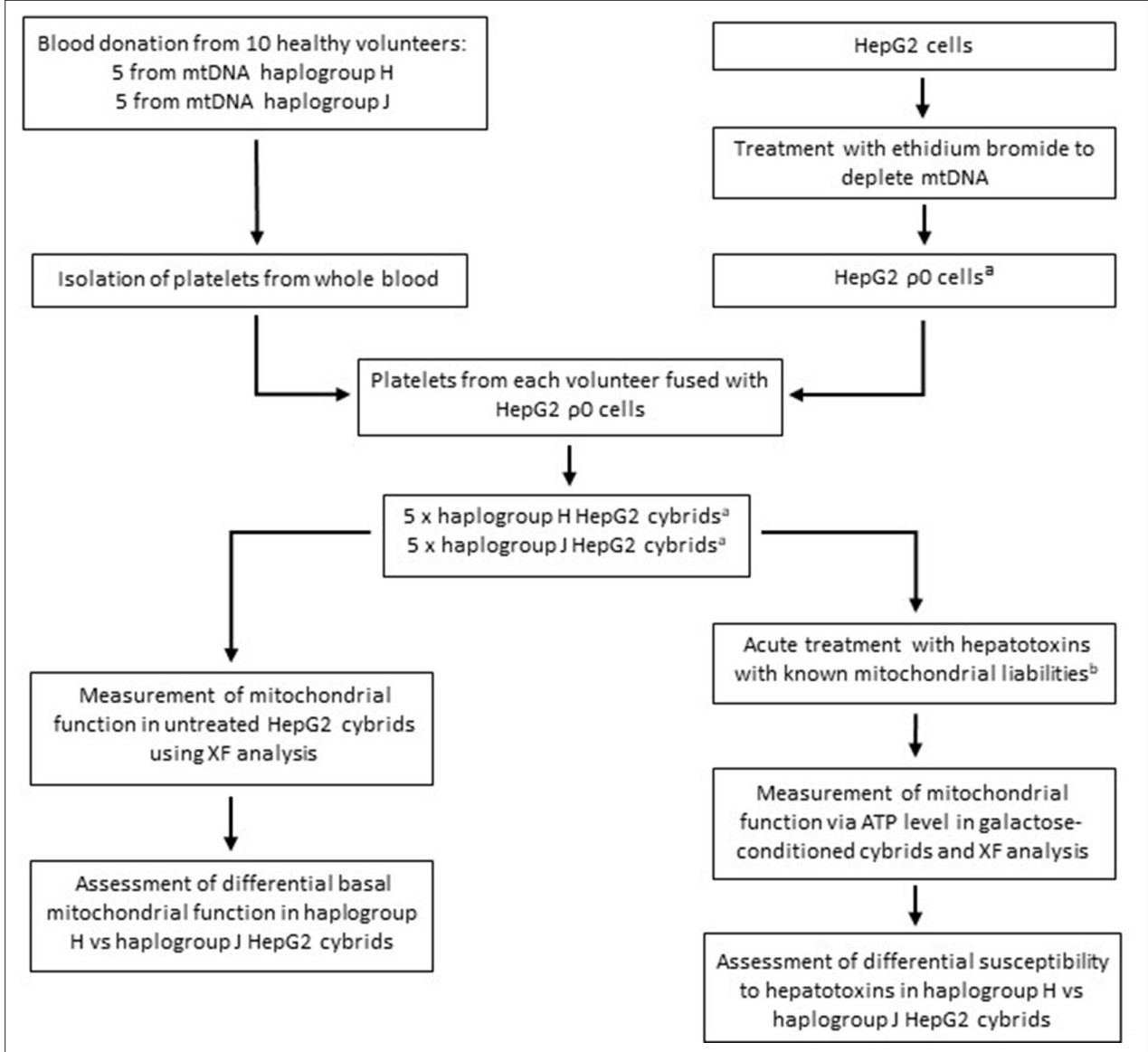

**Figure 1.** Study overview. [a] HepG2 $\rho 0$ cells were characterized to ensure the complete depletion of mtDNA and the expression of mtDNA-encoded proteins was confirmed in the HepG2 cybrids. Methods of characterization are described in the Supplementary information. [b] Flutamide, 2-hydroxyflutamide, and tolcapone, alongside non-hepatotoxic structural counterparts; bicalutamide and entacapone. Abbreviations: mtDNA, mitochondrial DNA; $\rho 0$, rho zero; XF, extracellular flux.

Whereas complex III – and IV-driven respiration was not significantly different between haplogroups H and J. Unlike complex I, complex II is entirely encoded in the nuclear genome, therefore, this difference in activity is unexpected. However, it has been noted that the introduction of the foreign mitochondrial genome into cybrids, can lead to the initiation of retrograde responses to the nucleus primarily via calcium signaling (*Luo et al., 1997*; *Amuthan et al., 2002*; *Srinivasan et al., 2016*). Such a regulatory feedback loop may not only lead to changes in the nuclear-encoded mitochondrial proteome but also potentially enhance the biogenesis of the respiratory machinery, which could explain this result. It should also be noted that these haplogroup-specific differences in complex activity are in contrast to overall oxidative phosphorylation parameters seen at basal (*Figure 2A*). This may arise due to technical differences in the methodology - complex activity is measured in an uncoupled system with access to excess fuel, as opposed to the coupled system for basal bioenergetic measurements. Alternatively, it may be that the rate of activity through the latter complexes (III and IV), which demonstrate no differences between haplogroups, is rate-limiting for basal oxidative

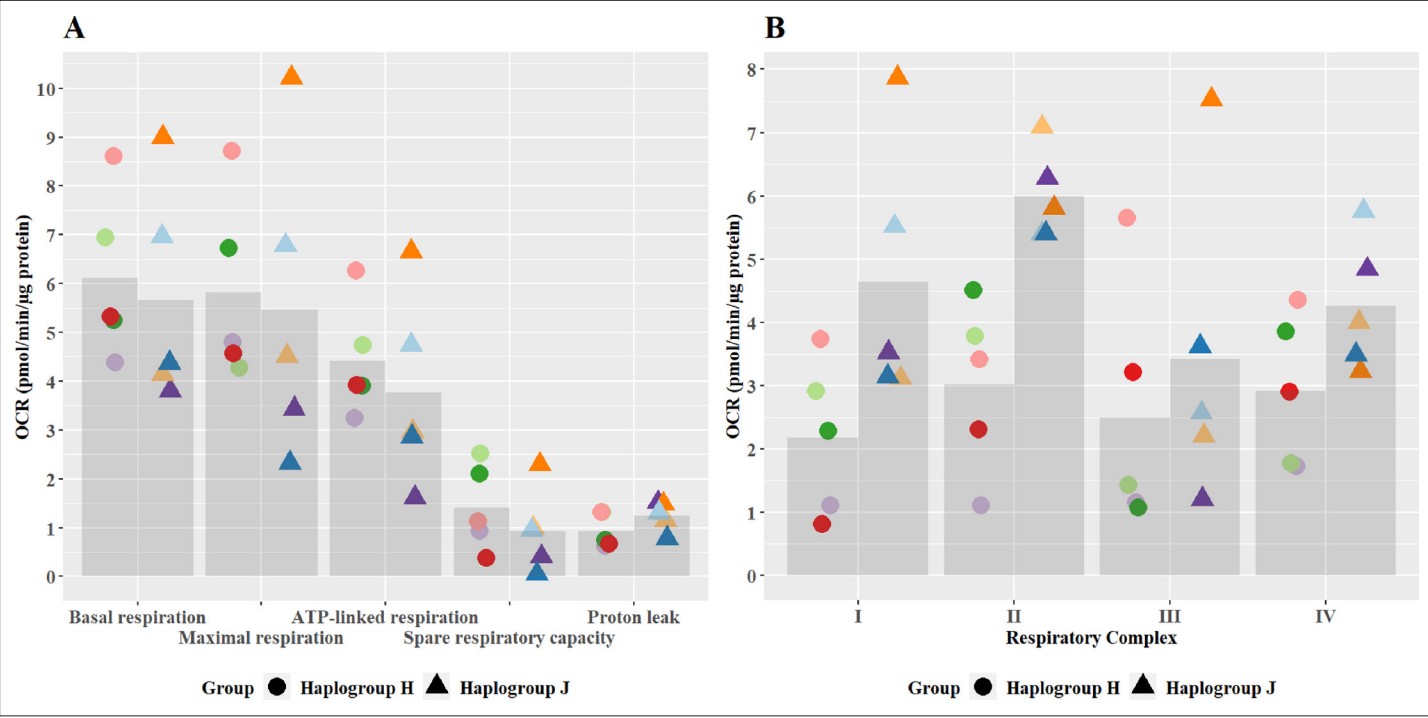

**Figure 2.** Basal mitochondrial function and respiratory complex activity in haplogroup H and J HepG2 cybrids. (**A**) Untreated haplogroup H and J cybrids were assessed using extracellular flux analysis and a mitochondrial stress test to measure: basal respiration, maximal respiration, ATP-linked respiration, spare respiratory capacity, and proton leak. (**B**) Untreated haplogroup H and J cybrids were assessed using extracellular flux analysis and respiration was stimulated by the supply of respiratory complex-specific substrates. Complex I-IV activity was defined as complex I-IV-driven maximal respiration. Data are presented from five haplogroup H cybrid cell lines and five haplogroup J cell lines (n=5 experiments were performed on each cybrid cell line). Each color point represents a single cybrid cell line. Abbreviations: OCR, oxygen consumption rate. Source data: *Figure 2—source data 1* file.xlsx. The results of all statistical tests can be viewed in *Supplementary file 2* – tab 2a.

The online version of this article includes the following source data for figure 2:

**Source data 1.** Source data of the basal mitochondrial function and respiratory complex activity of transmitochondrial cybrids displayed in *Figure 2*.

phosphorylation. However, whatever the reason, this phenomenon has been previously reported in the work of Strobbe et al, using transmitochondrial cybrids of haplogroups H, J, T, U, and K (*Strobbe et al., 2018*).

## Subtle haplogroup-specific changes in mitochondrial respiration were induced by treatment with flutamide

Flutamide was selected as a paradigm hepatotoxin that induces mitochondrial dysfunction via inhibition of the electron transport chain. Previous work has demonstrated that conditions of an acute metabolic switch, i.e., galactose-media, are required to circumvent the Crabtree effect and reveal the mitochondrial dysfunction induced by flutamide (*Ball et al., 2016*). Therefore, these experiments were performed in galactose media. As expected, a substantial, concentration-dependent decrease in ATP was evident when cybrids were treated with ≥33 M flutamide, however, this decrease was not significantly different between haplogroup H and J cybrids (*Figure 3A*). The effect on ATP was shown to be independent of cell death, as no significant lactate dehydrogenase [LDH] release occurred at any of the concentrations used (data not shown).

XF respirometric analysis (*Figure 3B–F*) revealed no differences between haplogroup H and haplogroup J in basal oxygen consumption (OCR), maximal OCR, and ATP-linked respiration in the presence of flutamide. Spare respiratory capacity (SRC), was greater in haplogroup H cybrids across the central concentrations of the dose-response curve, although this did not reach significance (J vs H: SRC mean difference –1.37 pmol/min/ug protein, 95% CI [–2.83, 0.09], p=0.063). Greater SRC is linked to protection against mitotoxicants, and, therefore, the larger SRC observed in haplogroup H may indicate a potential for haplogroup-specific protection against injury by flutamide. Furthermore,

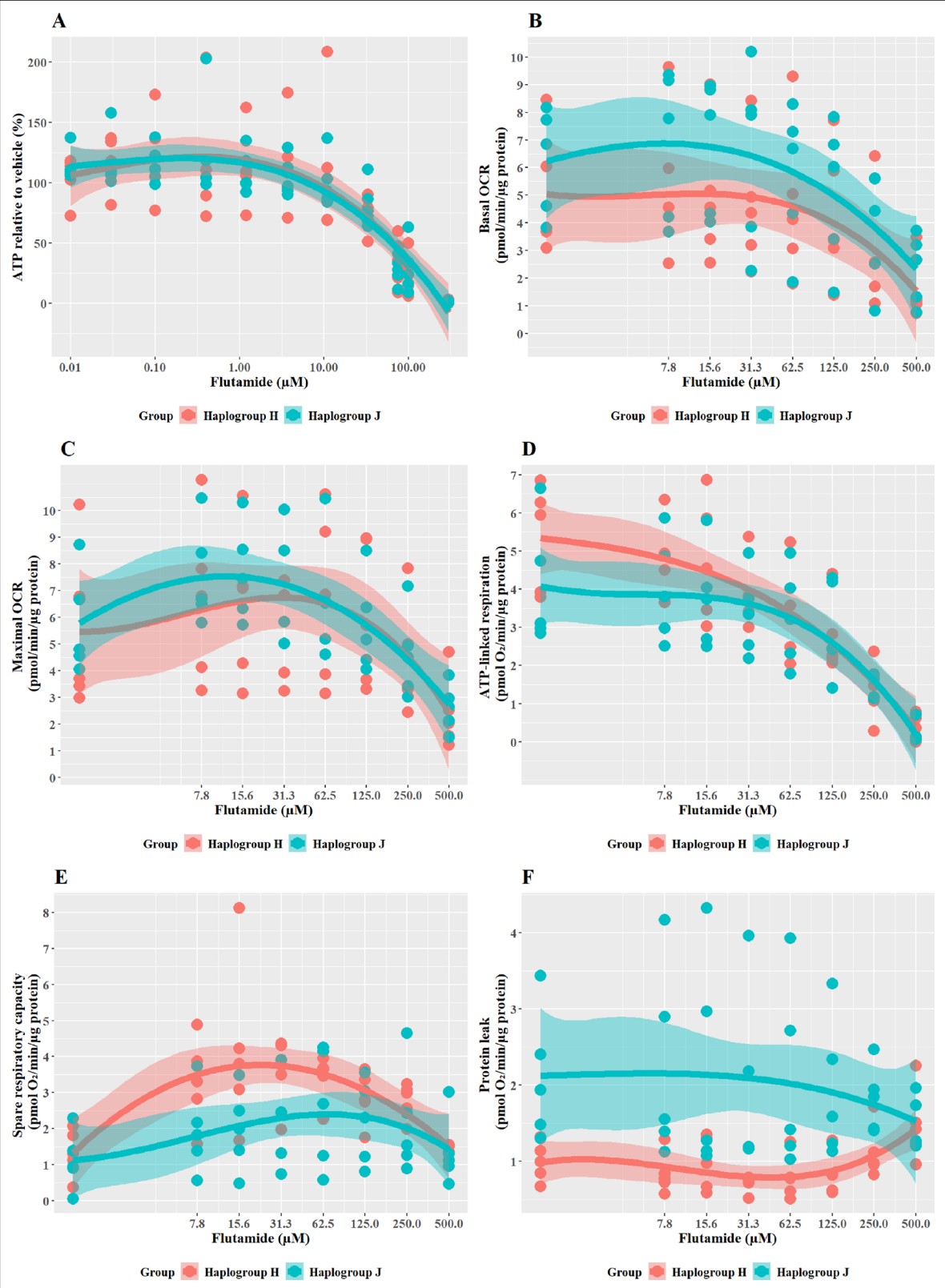

**Figure 3.** The effect of flutamide on ATP levels and mitochondrial respiratory function in haplogroup H and J HepG2 cybrids. (**A**) Cybrids were treated (2 hr) with up to 300 μM flutamide in a galactose medium. ATP values are expressed as a percentage of the vehicle control. (**B–F**) Changes in basal respiration, maximal respiration, ATP-linked respiration, spare respiratory capacity, and proton leak following acute treatment with flutamide (up to 500 μM). Data are presented from five haplogroup H cybrid cell lines and five haplogroup J cell lines (n=5 experiments were performed on each cybrid

*Figure 3 continued on next page*

Figure 3 continued

cell line). Shaded areas represent a 95% confidence interval of the fitted curve. Abbreviations: OCR, oxygen consumption rate. Source data: *Figure 3— source data 1* file.xslx. The results of all statistical tests can be viewed in *Supplementary file 2* – tab 2b.

The online version of this article includes the following source data for figure 3:

**Source data 1.** Source data of the effect of flutamide on ATP levels and mitochondrial respiratory function of cybrids displayed in *Figure 3*.

proton leak (PL), a negative indicator of mitochondrial health, was significantly greater in haplogroup J compared with haplogroup H (PL mean difference 1.29 pmol/min/ug protein, 95% CI [0.3, 2.27], p=0.016).

## Haplogroup J shows greater mitochondrial sensitivity to 2-Hydroxyflutamide than haplogroup H

2-hydroxyflutamide is the major circulating form of flutamide when it is administered clinically. When cybrids were treated with ≥33 M 2-hydroxyflutamide a substantial, concentration-dependent decrease in ATP level was evident. However, there was no significant difference in responses between haplogroup H and J cybrids (*Figure 4A*).

XF respirometric analysis (*Figure 4B–F*) revealed no difference between haplogroup H and J in basal OCR, maximal OCR, or proton leak. However, indicators of drug-induced mitochondrial toxicity, ATP-linked respiration, and SRC (*Figure 4D and E*), were both significantly lower in haplogroup J cybrids following exposure to 2-hydroxyflutamide (J vs H: ATP-linked respiration mean difference –1.13 pmol/min/ug protein, 95% CI [-2.02,–0.25], p=0.017; SRC mean difference –1.74 pmol/min/ug protein, 95% CI [-3.22,–0.25], p=0.027). This suggests that, at the level of oxidative phosphorylation, haplogroup J is more sensitive to the effects of 2-hydroxyflutamide, and/or haplogroup H is more protected. However, with both flutamide and 2-hydroxyflutamide, the drug-induced effects on bioenergetic parameters are not reflected in the results of cellular ATP content assays (*Figure 4A*), which showed no difference between haplogroups. In the ATP content assays, glycolytic ATP production is suppressed through the replacement of glucose with galactose as a fuel source to reveal the effects of any inhibition of oxidative phosphorylation (*Marroquin et al., 2007*; *Kamalian et al., 2015*). However, drug-induced changes in cellular ATP levels will reflect not only changes in oxidative phosphorylation but may also be affected by changes in the cellular consumption of ATP due to cell death or cell-protection pathways. Therefore, this demonstrates that extracellular flux analysis may provide greater sensitivity for detecting and quantifying comparative differences in mitochondrial dysfunction compared with measuring cellular ATP (nmoles/μg protein).

## Haplogroup J cybrids have greater respiratory complex activity but are more susceptible to inhibition of complex I and II activity by flutamide and 2-hydroxyflutamide

To further understand the cause of such differences in bioenergetic parameters, the effect of flutamide (complex I inhibitor) and 2-hydroxyflutamide (complex I and II inhibitors) on complex activity was measured across the cybrid cell lines (*Figure 5*). When treated with flutamide (*Figure 5A and B*), haplogroup J cybrids had significantly higher complex I activity than haplogroup H (J vs H: mean difference 2.15 pmol/min/ug protein, 95% CI [0.42, 3.89], p=0.021). However, despite an overall higher level of activity for complex I, haplogroup J cybrids showed a steeper dose-dependent reduction in their complex I activity by flutamide than haplogroup H. In line with its known targets, flutamide did not significantly reduce complex II activity in either haplogroup, however, there was generally a lesser impact on haplogroup J cybrids(J vs H: mean difference 1.23 pmol/min/ug protein, 95% CI [0.01, 2.47], p=0.051).

In line with the known mechanism, 2-hydroxyflutamide decreased the activity of complexes I and II in both haplogroups (*Figure 5C and D*). Haplogroup J cybrids again demonstrated a significantly greater maintenance complex I activity (J vs H: mean difference 2.81 pmol/min/ug protein, 95% CI [0.5, 5.13], p=0.023) and whilst complex II activity was generally greater in haplogroup J cybrids after treatment this did not reach significance (J vs H: mean difference 1.2 pmol/min/ug protein, 95% CI [–0.33, 2.73], p=0.111). Again, haplogroup J cybrids showed a steeper decline in activity from their higher baseline, suggesting a greater susceptibility to 2-hydroxyflutamide.

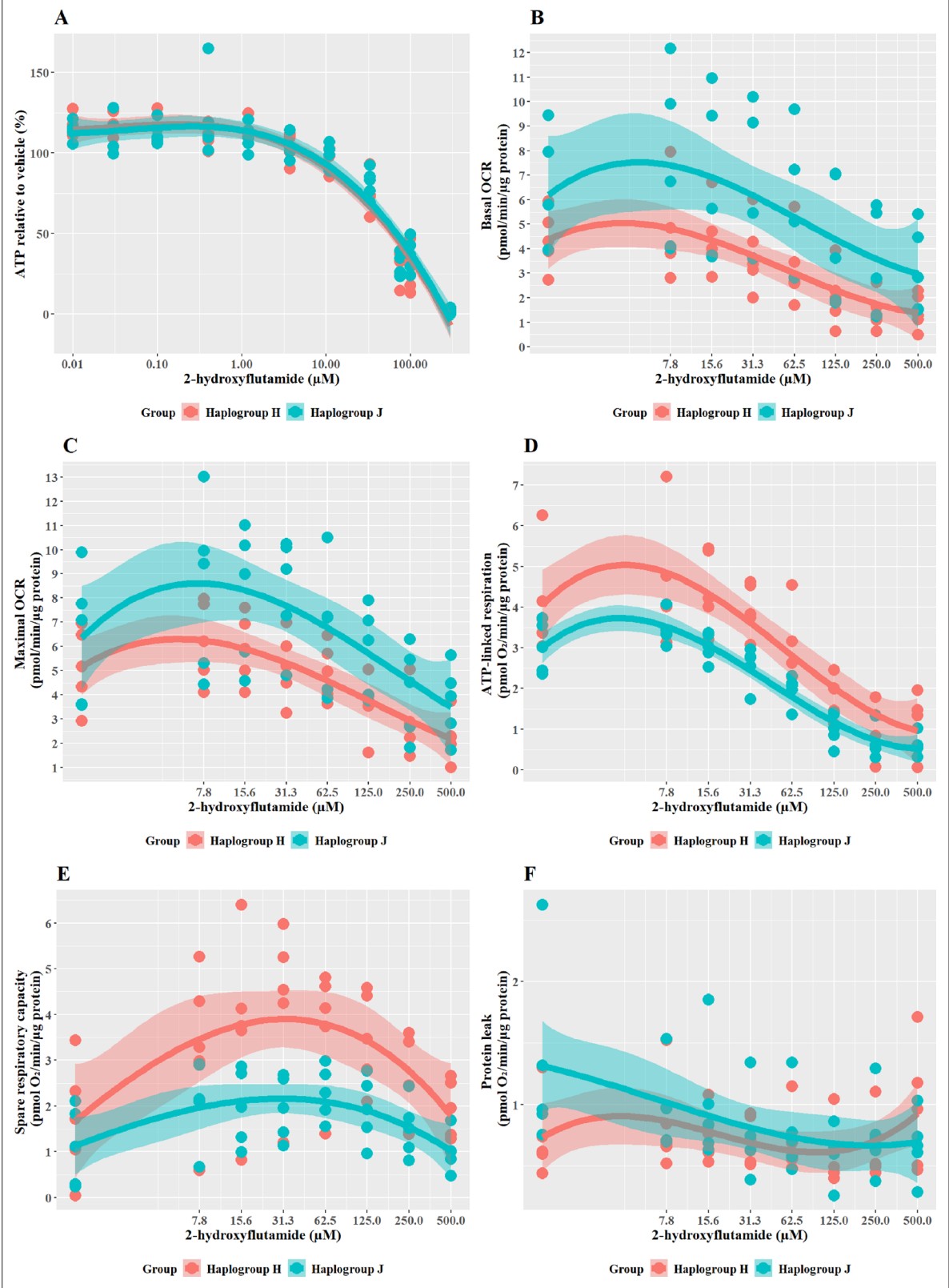

**Figure 4.** The effect of 2-hydroxyflutamide upon ATP levels and mitochondrial respiratory function in haplogroup H and J HepG2 cybrids. (**A**) Cybrids were treated (2 hr) with up to 300 μM 2-hydroxyflutamide in a galactose medium. ATP values are expressed as a percentage of those of the vehicle control. (**B–F**) Changes in basal respiration, maximal respiration, ATP-linked respiration, spare respiratory capacity, and proton leak following acute treatment with 2-hydroxyflutamide (up to 500 μM). Data are presented from five haplogroup H cybrid cell lines and five haplogroup J cell lines (n=5

*Figure 4 continued on next page*

*Figure 4 continued*

independent experiments were performed on each cybrid cell line). Shaded areas represent a 95% confidence interval of the fitted curve. Abbreviations: OCR, oxygen consumption rate. Source data: *Figure 4—source data 1* file.xslx. The results of all statistical tests can be viewed in *Supplementary file 2* – tab 2c.

The online version of this article includes the following source data for figure 4:

**Source data 1.** Source data of the effect of 2-hydroxyflutamide on ATP levels and mitochondrial respiratory function of cybrids displayed in *Figure 4*.

Although haplogroup J cybrids consistently showed higher complex I and II activity at basal (*Figure 2B*) and remained at a greater level than haplogroup H even post-treatment with flutamide and 2-hydroxyflutamide (*Figure 5A–D*), the relative reduction from basal levels were ultimately greater in haplogroup J than H. This supports the respirometry studies in whole cells (*Figures 3 and 4*) which suggested that haplogroup J was more susceptible to mitochondrial dysfunction induced by flutamide and 2-hydroxyflutamide and/or that haplogroup H is more resistant. This finding is in agreement with previous work using freshly isolated platelets from volunteers of known genotype, which also reported an increased sensitivity of haplogroup J to 2-hydroxyflutamide-induced inhibition of complex I-driven respiration (*Ball et al., 2021*).

## Haplogroup J cybrids are more susceptible to tolcapone-induced ATP depletion and mitochondrial dysfunction than haplogroup H cybrids

Tolcapone is a hepatotoxin that induces mitochondrial dysfunction via uncoupling of the electron transport chain. In the galactose test system, tolcapone induced a greater effect in haplogroup J than in haplogroup H cybrids. Specifically, cellular ATP levels were significantly lower in haplogroup J cybrids following treatment with tolcapone (J vs H: mean difference –29.46%, 95% CI [–50.83%, –8.09%], p=0.012) (*Figure 6A*). These effects on cellular ATP content were observed in the absence of cytotoxicity (i.e. no significant lactate dehydrogenase [LDH] release) at all the concentrations used (data not shown). This increased susceptibility of haplogroup J cybrids to tolcapone-induced depletion is in agreement with findings reported by *Ghelli et al., 2009*, in which a non-hepatic cybrid model was used to determine that haplogroup J (vs haplogroups H and U) was more susceptible to uncoupling by the neurotoxic metabolite, 2,5-hexanediol (*Ghelli et al., 2009*).

The pattern of changes to the parameters of oxidative phosphorylation that were induced by tolcapone is characteristic of its role as an uncoupler (*Kamalian et al., 2015*; *Figure 6B–F*). Specifically, basal respiration and proton leak showed a rapid dose-dependent increase, followed by decreases in ATP-linked respiration and SRC, before decreasing due to toxic effects. However, no differences were apparent between haplogroup H and J in any of the bioenergetic parameters measured. However, here again, it is noteworthy that there is a disconnect between the effect of haplogroup at the level of cellular ATP content and specific bioenergetic parameters but, in the case of tolcapone, this manifests in the opposite way to the findings for flutamide and 2-hydroxyflutamide, where haplogroup-specific effects were only evident in bioenergetic parameters and not cellular ATP content. This demonstrates that haplogroup-specific effects can vary across assays and are dependent upon the compound under evaluation and its mechanism of mitochondrial dysfunction. For example, tolcapone is a chemical uncoupler that binds and transports protons across the inner mitochondrial membrane by simple diffusion, in the same manner as carbonyl cyanide 4-(trifluoromethoxy) phenylhydrazone (FCCP). As such, the subtle inter-individual differences in bioenergetic phenotype between the cybrid cells may be immaterial. These results suggest that mitochondrial haplogroup may also influence other cellular pathways, post-uncoupling, that are not directly linked to oxidative phosphorylation, and it may be these that then lead to haplogroup-specific differences in cellular ATP content. Such pathways could include the induction of compensatory mechanisms for mitochondrial protection, which mitigate against reductions in cellular ATP such as an increase in mitochondrial mass, biogenesis, and changes to rates of mitochondrial fusion and fission (*Valera-Alberni and Canto, 2018*).

The results from this study offer important insights into the potential mechanisms underlying the onset of iDILI in certain individuals. In the case of tolcapone, four cases of liver failure among 100,000 individuals taking the drug, led to a black box warning and a switch to entacapone an alternative in-class compound. However, as entacapone has been reported to be less efficacious than tolcapone, the ability to make the drug selection based on the personalized risk of iDILI, would be of great

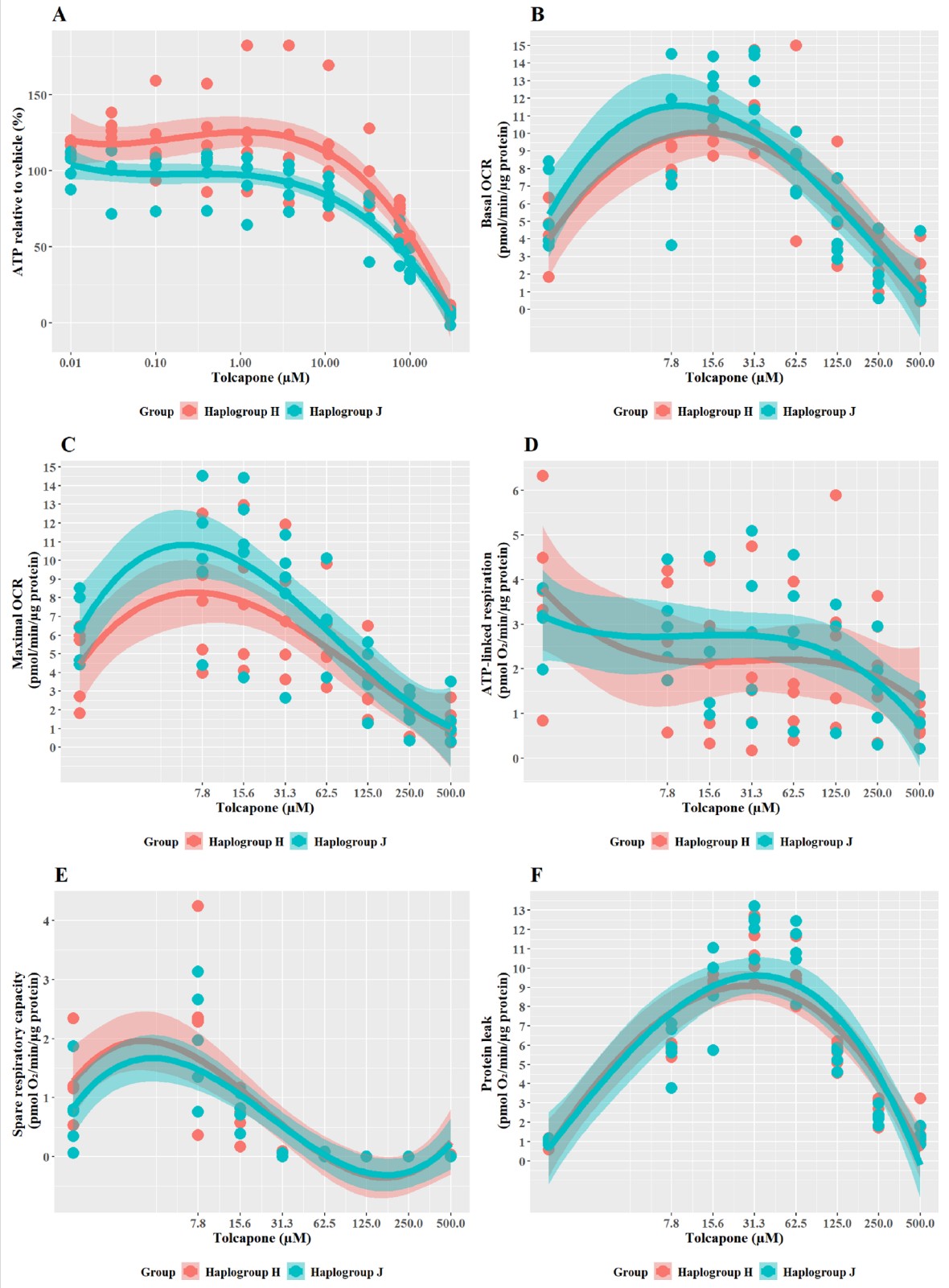

**Figure 5.** The effect of flutamide and 2-hydroxyflutamide upon respiratory complex I and II in haplogroup H and J HepG2 cybrids. Permeabilized cybrids were acutely treated with flutamide (**A, B**) or 2-hydroxyflutamide (**C, D**) (up to 250 μM) before a mitochondrial stress test using extracellular flux analysis. Complex I/II activity was defined as complex I/II-driven maximal respiration. Data are presented from five haplogroup H cybrid cell lines and

*Figure 5 continued on next page*

*Figure 5 continued*

five haplogroup J cell lines (n=5 experiments were performed on each cybrid cell line). Shaded areas represent a 95% confidence interval of the fitted curve. Source data: *Figure 5—source data 1* file.xslx. The results of all statistical tests can be viewed in *Supplementary file 2* – tab 2d.

The online version of this article includes the following source data for figure 5:

**Source data 1.** Source data of the effect of flutamide and 2-hydroxyflutamide on the activity of respiratory complex I and II in cybrids displayed in *Figure 5*.

value (*Rivest et al., 1999*; *Olanow, 2000*; *Watkins, 2000*; *Benabou and Waters, 2003*; *Olanow and Watkins, 2007*; *Lees, 2008*; *Longo et al., 2016*).

## Structurally related compounds, bicalutamide, and entacapone, are weaker hepatotoxins but display similar haplogroup-specific responses to their more potent counterparts

Cells were exposed to bicalutamide and entacapone as the non-hepatotoxic structural counterparts of flutamide and tolcapone, respectively to investigate whether haplogroup-specific differences are still evident. In line with previous work (*Ball et al., 2016*), results demonstrated that bicalutamide was less potent than flutamide or 2-hydroxyflutamide in terms of cellular ATP content (*Figure 7A*). Bicalutamide-treated cybrids of each haplogroup exhibited a similar decline in ATP content until the highest concentration (300 µM), with no significant differences observed between haplogroup H and J. No significant differences were evident between the two cybrid groups in parameters of mitochondrial function using XF analysis, however as was the case with flutamide and 2-hydroxyflutamide, haplogroup J exhibited higher proton leak in both control and treated cybrids (*Figure 7B–F*).

Entacapone induced a weaker decline in ATP levels compared to tolcapone, with a 50% decrease not reached at the highest concentrations tested (*Figure 8A*), reflective of previous work demonstrating it to be less toxic to mitochondria (*Kamalian et al., 2015*). However, there was no significant difference in ATP levels between the two haplogroups. Changes in bioenergetic parameters reflected the lower potency of entacapone as an uncoupler; dose-dependent increases in basal respiration and proton leak, accompanied by decreases in ATP-linked respiration and SRC, were evident only at higher test concentrations (*Figure 8B–E*). However, maximal OCR, and the linked parameter SRC, showed an increase at lower concentrations before ATP-linked respiration is decreased. This may represent a protective homeostatic response before uncoupling impacts the cellular ability to produce ATP via oxidative phosphorylation. Chemical-induced increases in SRC can indicate the early induction of protective mitochondrial response, driven by homeostatic mechanisms including increased fusion, increased access to fuels, or increased mitochondrial mass (*Marchetti et al., 2020*; *Valera-Alberni and Canto, 2018*). It is, therefore, of interest that haplogroup H cybrids induce a significantly greater increase in SRC activity (J vs H: mean difference –1.5 pmol/min/ug protein, 95% CI [-2.79,–0.21], p=0.027). This observation of potential early increased protection against entacapone-induced mitochondrial dysfunction in haplogroup H cybrids fits with the findings for tolcapone, where haplogroup J appeared to be more susceptible, perhaps due to a reduced protective response not evident at the concentrations of tolcapone tested.

## Overall discussion and conclusions

Research has begun to understand the role that mitochondrial DNA plays in defining an individual's susceptibility to adverse drug reactions (*Penman et al., 2020*; *Jones et al., 2021a*). However, limitations in clinical study design, in combination with conflicting findings, mean that the importance of mtDNA variation to toxicity remains unclear. Moreover, research using most cell models is complicated due to the presence of the nuclear genome. In our previous work using freshly isolated human platelets, to circumvent the influence of the nuclear genome, it was demonstrated that mitochondrial genotype-specific responses to drugs are present at the haplogroup level (*Ball et al., 2021*). These preliminary studies suggested that individual mitochondrial genotypes may play a role in influencing patient susceptibility to adverse drug reactions and indicate the need for further research. Such new knowledge is essential in the preclinical arena, to address the current lack of translatability between the identification of a compound with a mitochondrial liability and understanding its potential to induce toxicity in humans. Therefore, in this research, a panel of transmitochondrial cybrid cell lines

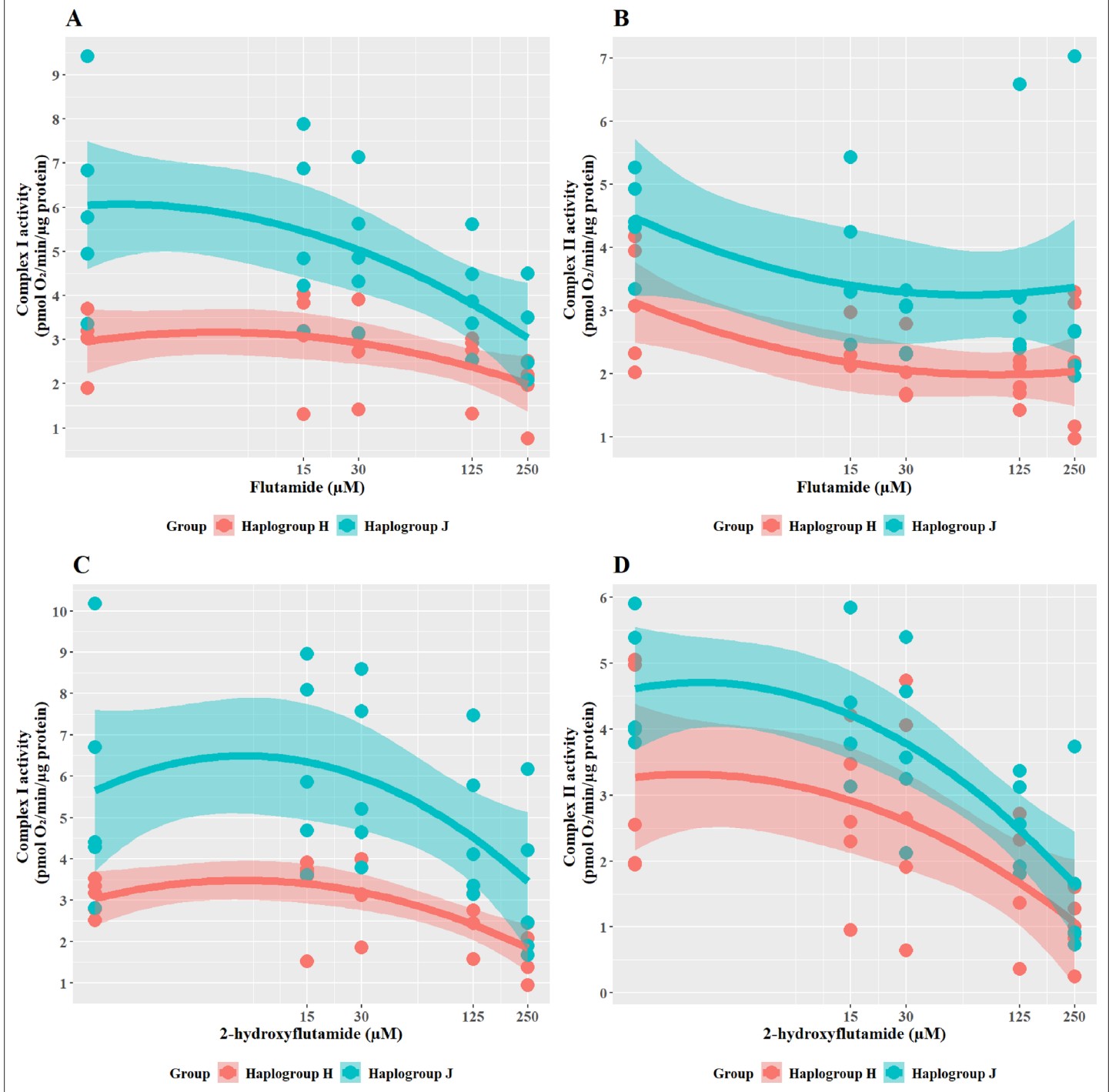

**Figure 6.** The effect of tolcapone on ATP levels and mitochondrial respiratory function in haplogroup H and J HepG2 cybrids. (**A**) Cybrids were treated (2 hr) with up to 500 µM tolcapone in a galactose medium. ATP values are expressed as a percentage of those of the vehicle control. (**B–F**) Changes in basal respiration, maximal respiration, ATP-linked respiration, spare respiratory capacity, and proton leak following acute treatment with tolcapone (up to 500 µM). Data are presented from five haplogroup H cybrid cell lines and five haplogroup J cell lines (n=5 experiments were performed on each cybrid cell line). Shaded areas represent a 95% confidence interval of the fitted curve Abbreviations: OCR, oxygen consumption rate. Source data: *Figure 6—source data 1* file.xslx. The results of all statistical tests can be viewed in *Supplementary file 2 – Table 2e*.

The online version of this article includes the following source data for figure 6:

**Source data 1.** Source data of the effect of tolcapone on ATP levels and mitochondrial respiratory function of cybrids displayed in *Figure 6*.

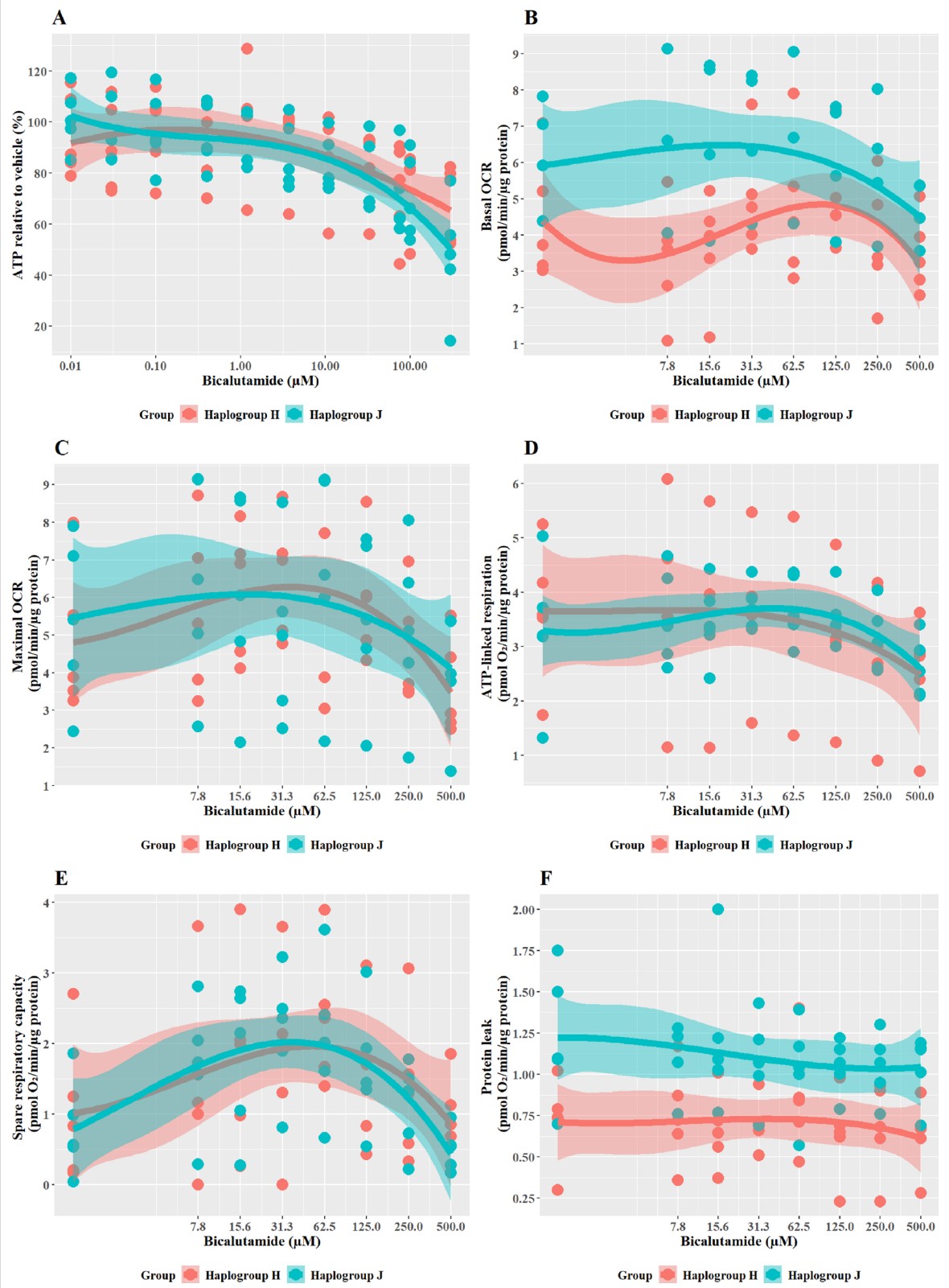

**Figure 7.** The effect of bicalutamide upon ATP levels and mitochondrial respiratory function in haplogroup H and J HepG2 cybrids. (**A**) Cybrids were treated (2 hr) with up to 500 µM bicalutamide in a galactose medium. ATP values are expressed as a percentage of those of the vehicle control. (**B–F**) Changes in basal respiration, maximal respiration, ATP-linked respiration, spare respiratory capacity, and proton leak following acute treatment with bicalutamide (up to 500 µM). Data are presented from five haplogroup H cybrid cell lines and five haplogroup J cell lines (n=5 independent

*Figure 7 continued*

experiments were performed on each cybrid cell line). Shaded areas represent a 95% confidence interval of the fitted curve. Abbreviations: OCR, oxygen consumption rate. Source data: *Figure 7—source data 1* file.xlsx. The results of all statistical tests can be viewed in *Supplementary file 2 – Table 2f*.

The online version of this article includes the following source data for figure 7:

**Source data 1.** Source data of the effect of bicalutamide on ATP levels and mitochondrial respiratory function of cybrids displayed in *Figure 7*.

was created as 'personalized models,' in which each cell line contains a distinct mitochondrial genotype against a stable HepG2 nuclear background. Importantly, these studies demonstrate for the first time that transmitochondrial cybrids can be created in HepG2 cells to contain the mitochondrial genotype of any individual of interest. In addition, they provide a practical and reproducible system to investigate the molecular and cellular consequences of variation in the mitochondrial genome. Thus this panel offers a novel, and unique in vitro model of idiosyncratic hepatotoxicity that will allow us to begin to map the functional effects of mitochondrial genotype. Due to their potential value to other researchers and in preclinical testing, we are happy to collaborate and share our cells and have plans to deposit them with a commercial vendor. However, a caveat should be noted, that the process of generating these cells, particularly the use of ethidium bromide and PEG, has the potential to introduce additional cellular changes in addition to the intended effects upon the mitochondrial genome. In mitigation, in this study all cells have been treated in the same way, minimizing inter-cell line variation. Furthermore, the advantage of using anucleate cells as mitochondrial donors removes the need for a further manipulation to chemically enucleate a donor cell.

This research provides new evidence for the role of mitochondrial genotype in the onset of idiosyncratic ADRs. It has shown that haplogroup J cybrids display an increased susceptibility to tolcapone-induced decrease in ATP and to mitochondrial dysfunction induced by 2-hydroxyflutamide, which supports previous findings on the increased sensitivity of haplogroup J to mitotoxicants (*Strobbe et al., 2018*). Importantly, it provides quantitative evidence that inter-individual variation in mitochondrial genotype can be a factor in determining sensitivity to mitochondrial toxicants. It should be noted that idiosyncratic responses in DILI are likely due to several small variations coming together to push an individual over the toxicity threshold. The fact that mitochondrial haplogroup could be one such small difference is very much worthy of note. Here, we have focussed on only two haplogroups, which are close phylogenetic relations. In addition, all platelet donors were healthy volunteers, with no known clinical diagnoses of mitochondrial disease. Therefore, this proof-of-concept paves the way for future studies examining differences across more phylogenetically diverse haplogroups and sub-haplogroups. One limitation of the current study is the high levels of variability encountered when grouping different sub-haplogroups as a major haplogroup population, i.e. H vs J. Therefore, additional studies should focus on creating several panels of cell lines of the same sub-haplogroups, each derived from a different individual, to gain insight into whether this variation has a genetic or experimental basis. Additionally, focused investigations could utilize platelets from individuals who have experienced iDILI, individuals with mitochondrial disease, or other potential susceptibility factors. Such studies may reveal more pronounced differences in the mitochondrial response to hepatotoxic drugs and strengthen the evidence for the importance of examining mtDNA in pharmacogenetic investigations. Future developments should also focus on adapting the cybrid model to cell types that can better recapitulate complex liver architecture and are potentially more metabolically competent e.g., HepaRG cells or iPSC-derived cells, to enhance in vivo applicability. It would also be of interest to test the amenability of cybrids to 3D spheroidal culture to further improve in vivo relevance (*Gaskell et al., 2016*; *Fang and Eglen, 2017*). However, in our study, the lack of CYP P450 enzymes was considered an advantage as we could separately study the effects of the parent drug, flutamide, and its major metabolite 2-hydroxyflutamide.

In conclusion, this study has established a novel, in vitro model that provides a preclinical representation of the interindividual variation underpinning iDILI, thereby offering much greater translatability to clinical scenarios compared with the current, homogenous preclinical models. These proof-of-principle studies have provided evidence that mitochondrial genotype is a factor determining individual susceptibility to hepatotoxins that target the mitochondria. Therefore, these established lines should be utilized for future in-depth characterization and testing of additional putative hepatotoxic compounds to shed further light on the impact of mitochondrial DNA variation on (idiosyncratic)

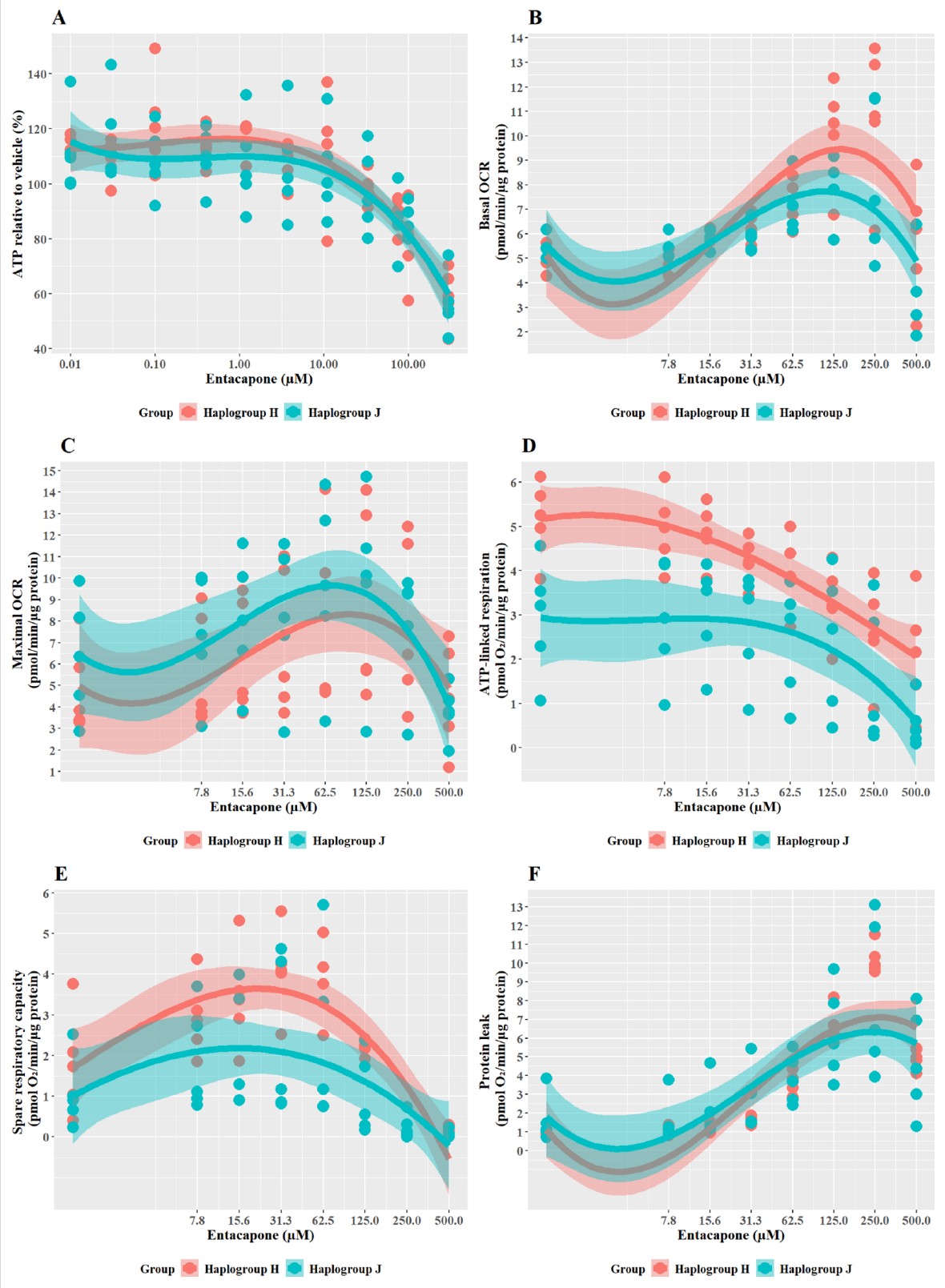

**Figure 8.** The effect of entacapone upon ATP levels and mitochondrial respiratory function in haplogroup H and J HepG2 transmitochondrial cybrids. (**A**) Cybrids were treated (2 hr) with up to 500 μM entacapone in a galactose medium. ATP values are expressed as a percentage of those of the vehicle control. (**B–F**) Changes in basal respiration, maximal respiration, ATP-linked respiration, spare respiratory capacity, and proton leak following acute treatment with entacapone (up to 500 μM). Data are presented from five haplogroup H cybrid cell lines and five haplogroup J cell lines (n=5

*Figure 8 continued on next page*

*Figure 8 continued*

independent experiments were performed on each cybrid cell line). Shaded areas represent a 95% confidence interval of the fitted curve. Abbreviations: OCR, oxygen consumption rate. Source data: *Figure 8—source data 1* file.xlsx. The results of all statistical tests can be viewed in *Supplementary file 2* – tab 2g.

The online version of this article includes the following source data for figure 8:

**Source data 1.** Source data of the effect of entacapone on ATP levels and mitochondrial respiratory function of cybrids displayed in *Figure 8*.

## Materials and methods

### Materials

All forms of DMEM were purchased from Life Technologies (Paisley, UK). HepG2 cells were purchased from the European Collection of Cell Cultures (Salisbury, UK). Cytotoxicity detection kits were purchased from Roche Diagnostics Ltd (West Sussex, UK). Clear and white 96-well plates were purchased from Fisher Scientific (Loughborough, UK) and Greiner Bio-One (Stonehouse, UK), respectively. All XF assay consumables were purchased from Agilent Technologies (CA, USA). All other reagents and chemicals were purchased from Sigma Aldrich (Dorset, UK) unless otherwise stated.

### Cohort

Ten healthy volunteers were recalled from the previously established HLA-typed archive to give blood, described in *Alfirevic et al., 2012*. These volunteers were selected based on their mitochondrial haplogroup (haplogroup H and J) and this genotyping has been described in our previous publication (*Ball et al., 2021*). Full information regarding haplogroup assignment based on characteristic SNP is available in *Supplementary file 1*. Mitochondrial subhaplogroups and SNPs of the mitochondrial DNA from each individual are described in the Supplementary Information. Ten donors were selected as an adequate number for this proof of principle study based on the feasibility of generating the transmitochondrial cybrids. The volunteers were eligible to take part in the study if they were aged between 18 and 60 years, healthy, and willing to donate one or more blood samples. The following exclusion criteria were applied and volunteers were not recruited if: they donated blood to transfusion services in the last 4 months; they had any medical problems (including asthma, diabetes, epilepsy, or anaemia), were on any medications, or if they had taken any recreational drugs in the last 6 weeks (including cannabis, speed, ecstasy, cocaine, and LSD). Women were excluded if pregnant. This project was approved by the North West of England Research Ethics Committee and all participants gave written informed consent. Volunteer confidentiality was maintained by double coding DNA samples and by restricting access to participants' data to trained clinical personnel. Detailed study eligibility and exclusion criteria have been published previously (*Alfirevic et al., 2012*).

### Generation of HepG2 cybrids

For a schematic representation of HepG2 cybrid generation see *Figure 9A*. HepG2 cells (ECACC Cat# 85011430, RRID: CVCL_0027) (≤passage 7) were cultured and passaged as required in HepG2 $\rho$0 cell medium (DMEM/F-12+GlutaMAX supplemented with FBS [10% v/v], L-glutamine [4 mM], sodium pyruvate [1 mM], HEPES [2 mM], and uridine [500 µM]) in the presence of ethidium bromide (EtBr; 1 µM). During this period, EtBr was removed for 48 hr every two weeks to help maintain cell viability. Following eight weeks' exposure, EtBr was removed from a subset of cells for one week before characterization to ensure cells were devoid of mtDNA i.e., were $\rho$0 cells. If a $\rho$0 cell phenotype was not evident, cells were returned to EtBr treatment at the same dose before removal for another week and re-testing. This was continued until a $\rho$0 cell population was observed. All cells were tested monthly to ensure an absence of mycoplasma contamination.

### Platelet isolation

Healthy volunteers, five from haplogroup H, and five from haplogroup J donated whole blood from which platelets were isolated according to methods previously described (*Ball et al., 2021*). Briefly,

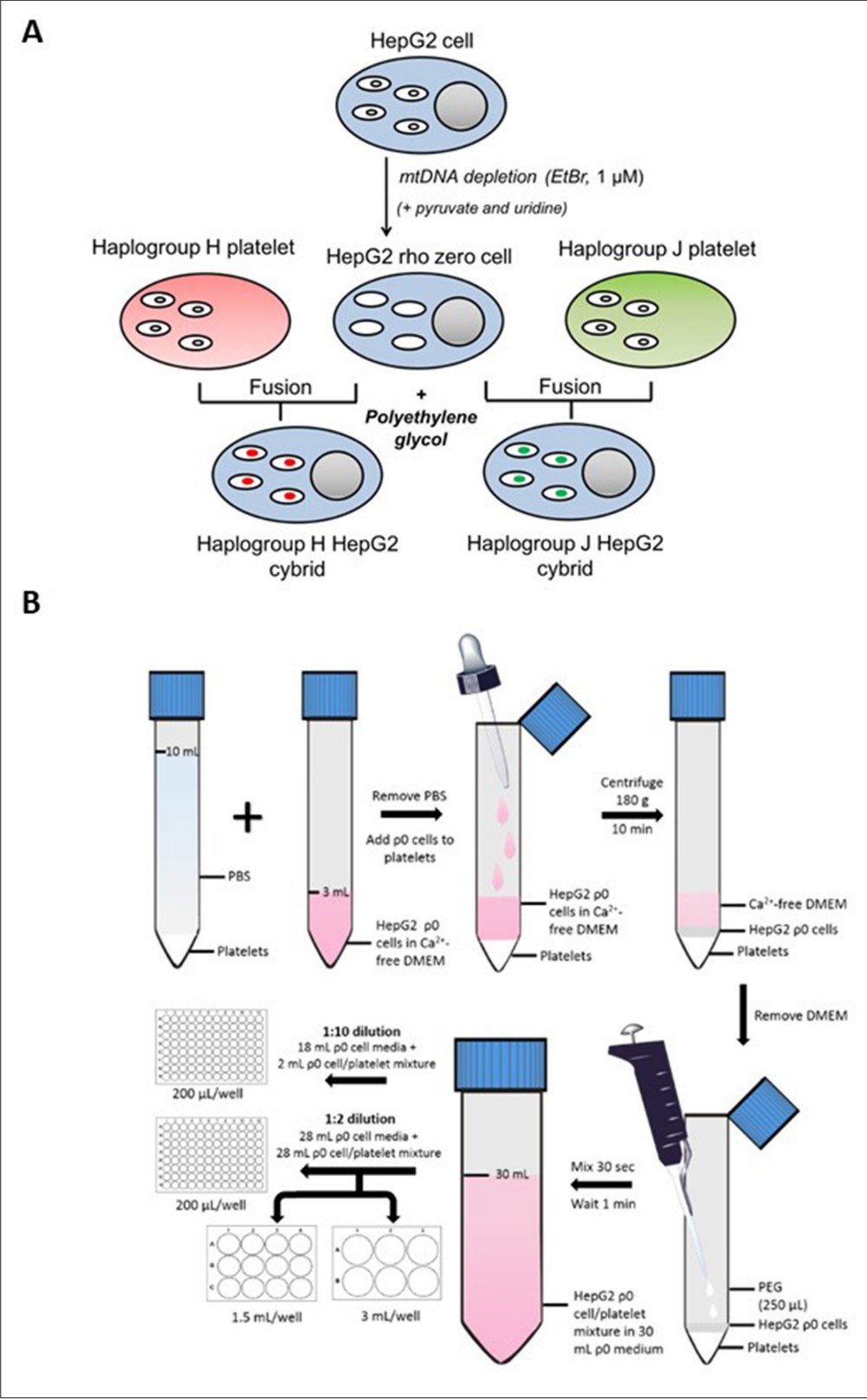

**Figure 9.** Generation of HepG2 transmitochondrial cybrids. (**A**) Schematic overview of HepG2 transmitochondrial cybrid generation. (**B**) Experimental procedure for the fusion of HepG2 $\rho$0 cells and platelets to generate HepG2 transmitochondrial cybrids.

*Figure 9 continued on next page*

*Figure 9 continued*

The online version of this article includes the following source data and figure supplement(s) for figure 9:

**Figure supplement 1.** Cell doubling time of HepG2 wild-type (WT) and HepG2 rho zero ( $\rho$ 0) cells.

**Figure supplement 1—source data 1.** Source data of the cell doubling time of cells displayed in *Figure 9—figure supplement 1*.

**Figure supplement 2.** Representative western blots of HepG2 wild-type (WT), rho zero, and cybrid cell lysates.

**Figure supplement 2—source data 1.** Table detailing the mitochondrial DNA (mtDNA) variation of the 10 healthy volunteers whose platelets were used to generate cybrids.

**Figure supplement 2—source data 2.** Table detailing the results of all statistical analysis for figures 2, 3, 4, 5, 6, 7 and 8.

50 mL of blood was donated by each volunteer, and this fresh whole blood was immediately processed by a series of density centrifugation steps to produce isolated platelets. Throughout the procedure, PGI$_2$ was used (1 µg/mL) to prevent platelet activation.

## Platelet fusion with HepG2 rho zero ( $\rho$ 0) cells

For an overview of the fusion experimental procedure see *Figure 9B*. During the final centrifugation step of platelet isolation, HepG2 $\rho$ 0 cells were collected by trypsinization and resuspended in HepG2 $\rho$ 0 cell medium. Following cell viability assessment (trypan blue; all viabilities were recorded at >90%), $\rho$ 0 cells (6 × 10$^6$) were centrifuged (1000 g, 5 min) and resuspended in Ca$^{2+}$-free DMEM (2 mL; supplemented with 1 µg/mL PGI2). This $\rho$ 0 cell suspension was overlaid onto isolated platelets using a Pasteur pipette to minimize disruption to the platelet pellet. The cell mixture was then centrifuged (180 g, no brake, 10 min) to form a multi-layered pellet of platelets and HepG2 $\rho$ 0 cells.

Following centrifugation, the supernatant was removed and polyethylene glycol (250 µL; PEG 50%) was added before resuspending the cell pellet (30 s) and incubating for 1 min. At the end of the incubation period, HepG2 $\rho$ 0 cell medium (30 mL) was added. A further 10-fold or twofold dilution with HepG2 $\rho$ 0 cell medium was performed before seeding into 96-well, 12-well, and 6-well plates.

Two days post-fusion, the media was replaced with a fresh HepG2 $\rho$ 0 cell medium. After a further two days, this was replaced with media consisting of equal volumes of HepG2 $\rho$ 0 medium and cybrid selection medium (high-glucose DMEM [glucose; 25 mM] supplemented with dialyzed FBS [10% v/v], amphotericin B [1.35 µM] and antibiotic/antimycotic solution [100 units penicillin/mL, 170 µM streptomycin and 270 nM amphotericin B]). Finally, two days later, the media was switched to a 100% cybrid selection medium. $\rho$ 0 cells are auxotrophic for pyruvate and uridine, so the absence of these two constituents was the basis for the selection of successfully fused cells i.e., HepG2 cybrids. Cells remaining after selection were characterized to ensure a HepG2 cybrid phenotype by measuring the expression and function of mtDNAencoded proteins (see *Figure 9—figure supplement 1 - 4* ). Cells were then cultured and passaged as required in a cybrid maintenance medium (DMEM high-glucose supplemented with FBS [10% v/v], L-glutamine [4 mM], sodium pyruvate [1 mM], and HEPES [1 mM]).

**Table 1.** Real-time PCR primers used to amplify regions of mitochondrial and nuclear DNA.

| Gene | Dye/probe | Additional information |
|---|---|---|
| RNase P (nDNA) | VICdye-labeled TAMRA probe | Location: chromosome 14, cytoband 14q11.2 |
| TERT (nDNA) | VICdye-labeled TAMRA probe | Location: chromosome 5, cytoband 5p15.33 |
| N/A Custom sequence (mtDNA) | FAMdye-labeled MGB probe | Oligonucleotide sequences: **hmito F5** CTTCTGGC CACAGCACTTAAAC **hmito R5** GCTGGTGTTAGG GTTCTTTGTTTT |
| ND-1 (mtDNA) | FAMdye-labeled MGB probe | Location: mtDNA 3307–4262 |

FAM = carboxyfluorescein. MGB = minor groove binder. ND-1 = NADH dehydrogenase-1. mtDNA = mitochondrial DNA. N/A = not applicable. TAMRA = 6-carboxytetramethyl-rhodamine. TERT = telomerase reverse transcriptase. VIC = 2'-chloro-7'phenyl-1,4-dichloro-6-carboxy-fluorescein.

## Confirmation of transmitochondrial cybrid status

### Cell doubling time

HepG2 wild-type (WT) cells and HepG2 rho zero ($\rho$0) cells were seeded at 30,000 cells/well in a 24-well plate in either $\rho$0 cell media (contains pyruvate and uridine) or selection media (devoid of pyruvate and uridine). On days 1, 3, 5, and 7 of culture, cells were collected by trypsinization and counted, following which the growth rate was calculated. Rho zero cells are auxotrophic for pyruvate and uridine, but WT cells are not, therefore, the absence of cell growth in selection media indicated the complete loss of mtDNA. For characterisation data see *Figure 9—figure supplement 1*.

### DNA extraction and real-time PCR

DNA extraction from HepG2 WT, HepG2 $\rho$0, and HepG2 cybrid cells was performed using a DNA mini kit (Qiagen, Manchester, UK) according to the manufacturer's instructions. Sample DNA concentrations and quality were then quantified using a Quant-iT PicoGreen dsDNA Assay Kit and nanodrop spectrophotometry, respectively (Fischer Scientific, Loughborough, UK).

Real-time PCR was carried out using two primers for regions of mtDNA; a custom sequence and ND-1 (complex I subunit) and two primers for regions of nuclear DNA; telomerase reverse transcriptase (TERT) and RNase P (Applied Biosystems, California, USA) (*Malik et al., 2011*). See *Table 1* for primer detail.

During sample preparation, 2 X Taqman genotyping master mix (5 µL), a nuclear DNA primer (0.5 µL), mtDNA primer (0.5 µL), dH$_2$0 (2 µL), and 10 ng DNA (2 µL) from each sample were combined to give a final sample concentration of 1 ng/µL in each well. Real-time PCR was then carried out using the viiA7 RT-PCR system (Life Technologies, UK). MtDNA copies per cell were calculated on the basis that each nuclear DNA primer was present in diploid copies per cell and used the following formula, where $x_1$=nuclear DNA primer cycle threshold ($C_t$) value, $x_2$=mtDNA primer $C_t$ value: mtDNA copies per cell = 2 ($2^{(x_1-x_2)}$) (*Schäfer, 2016*). For details of the results of these characterization experiments see *Table 2*.

### Detection of mitochondrial/nuclear DNA-encoded mitochondrial proteins

HepG2 $\rho$0, HepG2 WT, or HepG2 cybrid cells were lysed using sonication and 10 µg of lysate protein was resolved by sodium dodecyl sulphate-polyacrylamide *gel* electrophoresis (SDS-PAGE) using 4–12% Bis-Tris gel (Invitrogen, UK) in MOPS buffer (MOPS; tris-base; 1.21% w/v, sodium dodecyl sulphate; 0.20% w/v, EDTA; 0.06% w/v in distilled water (dH$_2$0)).

This gel was then transferred to a nitrocellulose membrane (GE Healthcare, Buckinghamshire, UK) in transfer buffer (tris-base; 0.30% w/v, glycine; 1.5% w/v, methanol; 20% v/v in dH$_2$0) and blocked using 10% non-fat dried milk in Tris Buffered Saline-Tween (TBS-T: TBS; 0.50% v/v, tween; 0.10% v/v in dH$_2$0).

Blocking solution was removed using TBS-T and the membrane was probed for CI-20, CII-30, CIII-core2, CIV-I, and CV-alpha subunits of complexes I-V of the electron transport chain using Mito-Profile Total OXPHOS Human WB Antibody Cocktail (RRID: AB_2533836, Cat No 45–8199, Abcam, Cambridgeshire, UK) (0.20% v/v in 10% non-fat dried milk in TBS-T). This was followed by anti-mouse secondary antibody (0.01% v/v in 10% non-fat dried milk in TBS-T) before visualization using an ECL system (GE Healthcare, Buckinghamshire, UK). For characterization data see *Figure 9—figure supplement 2*.

### Detection of electron transport chain function

Mitochondrial stress tests were performed on untreated HepG2 WT, p0, and HepG2 cybrid cells using extracellular flux analysis as described in the main text.

Extracellular flux analysis produced two raw outputs, oxygen consumption rate and extracellular acidification rate (ECAR). ECAR can be indicative of the glycolytic rate of the cell, however, it also takes into account changes in ECAR due to oxidative phosphorylation (*Mookerjee et al., 2015*). Therefore, the measure of PPR$_{gly}$, glycolytic production rate was used to quantify glycolysis, this was calculated by subtracting respiratory acidification contributions from the total proton production rate. The characterization data produced is in *Table 3*.

**Table 2.** Assessment of mtDNA content in HepG2 wild-type (WT), HepG2 rho zero ($\rho$0), and HepG2 cybrids.

*Custom mtDNA sequence: Oligonucleotide sequences: hmito F5 CTTCTGGCCACAGCACTTAAAC, hmito R5 GCTGGTGTAGGGTTCTTTGTTTT. The Ct value of both mtDNA primers increased dramatically in $\rho$0 compared to WT cells whilst nuclear DNA Ct values remained consistent across all three cell lines. Similarly, both HepG2 WT and cybrid cells had thousands of mtDNA copies/cell in contrast to the $\rho$0 cells which had less than one copy/cell. Abbreviations: $C_t$, cycle threshold; ND-1, NADH dehydrogenase-1; RNase, ribonuclease P RNA component H1; TERT, telomerase reverse transcriptase; WT, wild-type; $\rho$0, rho zero. $C_t$ values are displayed as the mean (SEM) of n=3 experiments.

| | Primer $C_t$ value | | | | mtDNA copies/cell | | | |
|---|---|---|---|---|---|---|---|---|
| Sample | TERT | RNase P | ND-1 | Custom* | TERT/ ND-1 | TERT/ custom | RNase P/ND-1 | RNase P/custom |
| HepG2 WT | 30.6 (1.00) | 31.3 (1.11) | 18.5 (0.120) | 18.9 (0.140) | 9280 | 6985 | 14563 | 10960 |
| HepG2 ρ0 | 27.7 (0.0600) | 27.2 (0.0700) | 34.7 (0.740) | 32.0 (0.180) | 0.00400 | 0.0240 | 0.00300 | 0.0170 |
| HepG2 cybrid | 29.8 (1.10) | 30.4 (0.550) | 18.9 (0.730) | 19.0 (0.0900) | 3875 | 3590 | 5673 | 5256 |

**Table 3.** Differences in parameters of mitochondrial function in HepG2 wild-type (WT), rho zero ($\rho$ 0), and cybrid cells.

HepG2 WT and cybrid cells exhibited a classic response to the series of mitochondrial inhibitors used to perform the mitochondrial stress test whereas the $\rho$ 0 cells did not respond to these inhibitors and had very low basal OCR, all of which was due to non-mitochondrial respiration. The PPR$_{gly}$ (used to quantify glycolysis)/OCR ratio was also higher in WT and $\rho$ 0 cells compared with cybrid cells. Abbreviations: OCR, oxygen consumption rate; PPR$_{gly}$, proton production rate attributed to glycolysis; WT, wild-type. Values are displayed as mean (SEM) of n=3 experiments. Source data: *Table 3*, *Table 3—source data 1* file.xslx

| HepG2 cell type | Basal OCR (pmol/min/µg protein) | PPR$_{gly}$/OCR | % Non-mitochondrial respiration |
|---|---|---|---|
| WT | 6.52 (0.350) | 0.354 (0.0900) | 28.4 (0.310) |
| Rho zero | 0.450 (0.100) | 55.5 (0.930) | 102 (4.33) |
| Cybrid | 6.53 (0.400) | 0.466 (0.660) | 24.3 (0.510) |

The online version of this article includes the following source data for table 3:

**Source data 1.** Source data of the bioenergetic analysis of cells displayed in *Table 3*.

$$PPR_{gly} = PPR_{tot} - PPR_{resp}$$

$$where$$

$$PPR_{tot} = \frac{ECAR}{BP} \ and \ PPR_{resp} = \left( \frac{10^{pH} - 6.093}{1 + 10^{pH} - 6.093} \right) \left( \frac{max \ H^+}{o_2} \right) (OCR_{tot} - OCR_{rot})$$

## Equations for the calculation of PPR$_{gly}$ from mitochondrial stress tests

Abbreviations: PPR$_{gly}$, proton production rate attributed to glycolysis; PPR$_{resp}$, proton production rate attributed to respiration; PPR$_{tot}$, total proton production rate; ECAR, extracellular acidification rate; BP, buffering power; max $H^+/O_2$, derived acidification for the metabolic transformation of glucose oxidation; OCR$_{tot}$, total oxygen consumption rate; OCR$_{rot}$, oxygen consumption rate following rotenone injection (*Kelly, 2018*).

## Assessment of mitochondrial function at basal state and following incubation with flutamide, 2-hydroxyflutamide, and tolcapone

Dual assessment of mitochondrial function (ATP content) alongside cytotoxicity (LDH release)

### Cell and reagent preparation

HepG2 cybrids were collected by trypsinization and seeded on a collagen-coated flat-bottomed 96-well plate in cybrid maintenance medium (20,000 cells/50 µL/well) and incubated (24 hr, 37 °C, 5% CO$_2$). Cells were then washed three times in serum-free galactose medium (DMEM containing 10 mM galactose and 6 mM L-glutamine) before the addition of galactose medium (50 µL) and further incubation (2 hr, 37 °C, 5% CO$_2$). This acute metabolic modification is sufficient to allow the identification of drugs that induce mitochondrial dysfunction, by reducing the ATP yield from glycolysis, thereby increasing reliance on OXPHOS for ATP production. Flutamide, 2-hydroxyflutamide, tolcapone, bicalutamide, and entacapone were each serially diluted to generate a concentration range of 0.01–300 µM in galactose medium. Diluted compounds (50 µL) were then added to each well (total well volume; 100 µL) and cells were incubated (2 hr, 37 °C, 5% CO$_2$) before conducting assays to assess mitochondrial function and cytotoxicity. All assays used ≤0.5% DMSO as vehicle control.

### ATP content assay

ATP content was assessed by the addition of cell lysate (10 µL) and ATP standard curve solutions to a white-walled 96-well plate. ATP assay mix (40 µL; prepared according to the manufacturer's instructions) was then added and bioluminescence was measured (Varioskan, Thermo Scientific).

## LDH release assay

LDH release was determined by the assessment of 25 L supernatant and 10 L cell lysate from each well, before measurement, using the cytotoxicity detection kit and reading at 490 nm. LDH release was calculated as LDH supernatant/(supernatant + lysate).

## Normalization (BCA assay)

Protein content was determined using cell lysate (10 µL) and protein standards (10 µL). BCA assay fluorescence was then measured at 570 nm.

## Extracellular flux analysis

HepG2 cybrids were collected by trypsinization and seeded on a collagen-coated XFe96 cell culture microplate (25,000 cells/100 µL medium/well; 96-well plate) and incubated (37 °C, 5% $CO_2$) overnight.

## Mitochondrial stress test

Please refer to *Ball et al., 2016* for a detailed description of this method (*Ball et al., 2016*). Briefly, cells were incubated (1 hr, 37 °C, 0% $CO_2$) before the replacement of culture medium with 175 µL of unbuffered Seahorse XF base medium supplemented with glucose (25 mM), L-glutamine (2 mM), sodium pyruvate (1 mM), pre-warmed to 37 °C (pH 7.4). Following an equilibration period, measurements were taken to establish a basal oxygen consumption rate (OCR) before the acute injection of each of the five test compounds (7.8–500 µM). Following compound injection, a mitochondrial stress test consisting, of sequential injections of oligomycin (ATP synthase inhibitor; 1 µM), FCCP (uncoupler; 0.5 µM), and rotenone/antimycin A (complex I/III inhibitors, respectively; 1 µM each), was performed.

## Respiratory complex assays

Please refer to *Salabei et al., 2014* for a detailed description of this method (*Salabei et al., 2014*). Briefly, the culture medium was replaced with mitochondrial assay solution buffer (MAS: $MgCl_2$; 5 mM, mannitol; 220 mM, sucrose; 70 mM, $KH_2PO_4$; 10 mM, HEPES; 2 mM, EGTA; 1 mM, BSA; 0.4% w/v) containing constituents to permeabilize cells and stimulate oxygen consumption via complex I (ADP; 4.6 mM, malic acid; 30 mM, glutamic acid; 22 mM, BSA; 30 µM, PMP; 1 nM), complex II (ADP; 4.6 mM, succinic acid; 20 mM, rotenone; 1 µM, BSA; 30 µM, PMP; 1 nM), complex III (ADP; 4.6 mM, duroquinol; 500 µM, rotenone; 1 µM, malonic acid; 40 µM, BSA; 0.2% w/v, PMP; 1 nM), or complex IV (ADP; 4.6 mM, ascorbic acid; 20 mM, TMPD (N, N, N', N'-tetramethyl-p-phenylenediamine); 0.5 mM, antimycin A; 2 mM, BSA; 30 mM, PMP; 1 nM). Following a basal measurement (no equilibration period) of three cycles of mix (30 s), wait (30 s) and measure (2 min), flutamide or 2-hydroxyflutamide (or MAS buffer for determination of basal complex activity) was injected followed by a mitochondrial stress test as detailed previously, only each measurement cycle was 3 min rather than 6 min.

## Statistical analysis

In total, 10 distinct cybrid cell lines (five distinct haplogroup H cell lines, and five distinct haplogroup J cell lines) were generated, one from each recruited volunteer. Each population was tested as an independent experiment (n=1), therefore, giving a total of n=5 for each cybrid cell line/volunteer. Each independent experiment contained a minimum of three technical replicates.

Platelets were provided to the investigator blinded to haplogroup, to avoid bias; therefore, cybrid generation, experiments, and subsequent data analyses were performed on cybrids for which the haplogroup was unknown. Unblinding occurred at the stage at which datasets were combined to enable the subsequent statistical comparison of haplogroup H vs haplogroup J.

Parameters for comparison were predefined to discourage statistical bias during analyses. These were, for both the assessment of mitochondrial function and drug-induced mitochondrial dysfunction: basal, maximum, and ATP-linked respiration, spare respiratory capacity, proton leak, and complex I-IV activity.

All statistical analysis was conducted in R version 4.2.0. Each dataset was analyzed using a linear mixed model with the subject fitted as a random effect and the haplogroup fitted as a fixed effect. The dose-response relationship within each haplogroup was modeled using a polynomial to degree 3 and fitted to the log10 concentration. The untreated group was also included and given a concentration of log10(1). The exception to this was in the model fitted to assess the effect of flutamide

and 2-hydroxyflutamide upon respiratory complex I and II where a polynomial to degree 2 was used as the experiments only contained five concentrations. Post-hoc tests were then obtained from the fitted models to compare the mean response between the two haplogroups. For the models with the fitted polynomials, this comparison was made at the mean concentration i.e., in the middle of the fitted dose-response curve. The results of all statistical tests can be viewed in *Supplementary file 2*.

## Acknowledgements

The authors would like to thank the Royal Liverpool Research Facility, in particular, Lisa Gaskell, for the recruitment of volunteers and sample collection, and Prof. Dr. Peter Seibel and colleagues for their assistance in the generation of rho zero cells. This work was supported by the Centre for Drug Safety Science supported by the Medical Research Council, United Kingdom (Grant Number G0700654); and GlaxoSmithKline as part of an MRC-CASE studentship (grant number MR/L006758/1).

## Additional information

### Competing interests

Amy Louise Ball, Ana Alfirevic: holds a grant from GlaxoSmithKline (GSK) for this work. The author has no other competing interests to declare. Carol E Jolly: salary funded by Janssen Pharmaceutical, paid to University of Liverpool. The author has no other competing interests to declare. Mark G Lennon: is an employee of GlaxoSmithKline (GSK). Jonathan J Lyon: is an employee of GlaxoSmithKline (GSK). Unpaid role as a member of the Investigative Toxicology Leaders Forum (ITLF) representing GSK on this group. Provides unpaid consultation to Cambridge University in areas of drug development. The author has no other competing interests to declare. Amy E Chadwick: holds a grant from GlaxoSmith-Kline (GSK) for this work and received other funding from Janssen in 2017-2020 (paid directly to University of Liverpool). The author has no other competing interests to declare.

### Funding

| Funder | Grant reference number | Author |
| --- | --- | --- |
| Medical Research Council | Centre for Drug Safety Science G0700654 | Carol E Jolly |
| GlaxoSmithKline | MRC-CASE studentship grant number MR/L006758/1 | Amy Louise Ball |

The funders had no role in study design, data collection and interpretation, or the decision to submit the work for publication.

### Author contributions

Amy Louise Ball, Conceptualization, Formal analysis, Investigation, Visualization, Methodology, Writing – original draft, Project administration; Carol E Jolly, Investigation, Methodology, Writing – review and editing; Mark G Lennon, Formal analysis, Writing – review and editing; Jonathan J Lyon, Supervision, Funding acquisition, Investigation, Writing – review and editing; Ana Alfirevic, Supervision, Funding acquisition, Project administration, Writing – review and editing; Amy E Chadwick, Conceptualization, Resources, Data curation, Supervision, Investigation, Methodology, Project administration, Writing – review and editing

### Author ORCIDs

Amy E Chadwick (iD) http://orcid.org/0000-0002-7399-8655

### Ethics

Human subjects: This work with human material in this project was approved by the North West of England Research Ethics Committee and all participants gave written informed consent and consent to publish. All procedures were in accordance with the ethical standards of the North West of England Research Ethics Committee (Cell Archive of HLA Typed Healthy Volunteers (HLA), CRN ID 7787,

IRAS ID: 15623) with the 1964 Helsinki declaration and its later amendments or comparable ethical standards.

## Decision letter and Author response
Decision letter https://doi.org/10.7554/eLife.78187.sa1
Author response https://doi.org/10.7554/eLife.78187.sa2

---

# Additional files

## Supplementary files
• Supplementary file 1. Table detailing the mitochondrial DNA (mtDNA) variation of the 10 healthy volunteers whose platelets were used to generate cybrids. Full details of the bioinformatics analysis and haplogroup assignment of these donors can be found in *Ball et al., 2021*. Briefly, variant calling was performed using the Genome Analysis Toolkit (GATK; v3.2.2) before these results were input into HaploGrep2 (v2.1.0) for the determination of mtDNA haplogroup. Additional SNPs refer to SNPs that were present in the sample but were not characteristic of the assigned haplogroup. Abbreviations: SNP, single nucleotide polymorphism.

• Supplementary file 2. Results of statistical analysis of the effect of haplogroup on drug-induced changes in bioenergetic parameters in an excel spreadsheet format. Each dataset was analyzed using a linear mixed model with the subject fitted as a random effect and the haplogroup fitted as a fixed effect as described in the statistical analysis section. Key to the data presented in each tab: 2 a - accompanying analysis for *Figure 2*; Basal mitochondrial function and respiratory complex activity in haplogroup H and J HepG2 cybrids. 2b - accompanying analysis for *Figure 3*; The effect of flutamide on ATP levels and mitochondrial respiratory function in haplogroup H and J HepG2 cybrids. 2 c - accompanying analysis for *Figure 4*; The effect of 2-hydroxyflutamide upon ATP levels and mitochondrial respiratory function in haplogroup H and J HepG2 cybrids. 2d - accompanying analysis for *Figure 5*; The effect of flutamide and 2-hydroxyflutamide upon respiratory complex I and II in haplogroup H and J HepG2 cybrids. 2e - accompanying analysis for *Figure 6*; The effect of tolcapone on ATP levels and mitochondrial respiratory function in haplogroup H and J HepG2 cybrids. 2 f - accompanying analysis for *Figure 7*; The effect of bicalutamide upon ATP levels and mitochondrial respiratory function in haplogroup H and J HepG2 cybrids. 2 g - accompanying analysis for *Figure 8*; *Figure 8* The effect of entacapone upon ATP levels and mitochondrial respiratory function in haplogroup H and J HepG2 transmitochondrial cybrids. * denotes unit of %.

• Transparent reporting form

## Data availability
Source Data files have been provided for Figures 2, 3, 4, 5, 6 and supplementary figures s1, s3, s4, s5, s6.

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
