## [Editor Report]

The aim of this study was to demonstrate the role of varying mitochondrial DNA levels as an important factor in drug-induced cell injury, hence creating a novel in vitro model which was representative of the diversity in mitochondrial genotype. The authors take a clever approach by using cybrid cell lines to test the role of mtDNA variations, both mutational load, and DNA level, and propose that such cell models could be potentially representative of mitochondrial genome diversity. Their findings provide evidence that these mechanisms could play a role in individual susceptibility to hepatic adverse drug reactions. This also adds an important understanding of the role of mitochondria in the onset of drug-induced toxicity.

---

## [Decision Letter]

**Decision letter after peer review:**

[Editors’ note: the authors submitted for reconsideration following the decision after peer review. What follows is the decision letter after the first round of review.]

Thank you for submitting the paper "The generation of HepG2 transmitochondrial cybrids to reveal the role of mitochondrial genotype in idiosyncratic drug-induced liver injury: a translational in vitro study" for consideration by *eLife*. Your article has been reviewed by 2 peer reviewers, and the evaluation has been overseen by a Reviewing Editor and a Senior Editor. The reviewers have opted to remain anonymous.

Comments to the Authors:

We are sorry to say that, after consultation with the reviewers, we have decided that this work will not be considered further for publication by *eLife*.

In this manuscript, Amy Ball et al., use HepG2 cell-derived mitochondrial Cybrids to study the effects of drug-induced liver injury. The aim here is to demonstrate the role of varying mitochondrial DNA levels as an important factor in drug-induced cell injury. The authors use Cybrid cell lines derived from Rho zero HepG2 Cells and H- and J haplotypes of blood cells from human donors. The use of cybrid cell lines to test the role of mtDNA variations, both mutational load, and DNA level, is an excellent idea. However, this manuscript contains many serious defects in terms of concept and experimental approach.

Weaknesses: A major weakness is the use of HepG2 cells which is derived from a human hepatic tumor. Its genotype and metabolic profile are substantially altered from the normal hepatic cells. As the authors noted, one example is the absence of many inducible microsomal CYPs. The effects of Hydroxyflutamide, the metabolic product on mitochondrial function are not the same as the effects of flutamide hydroxylated in cells by the action of endogenously expressed CYPs.

Another important drawback is that the analysis is strictly dependent on respiratory flux analysis. It is not clear why mtDNA contents are not analyzed. To that matter it is not clear if the mtDNA contents in donor haplotypes H and J are indeed different and how much different. Using two haplogroups with unknown mtDNA contents is not sufficient. One needs to generate cell lines containing multiple levels of mtDNA under the same nuclear DNA background.

*Reviewer #1 (Recommendations for the authors):*

Line 87: Is haplogroup J more strongly linked to iDILI than haplogroup H? If so, please note and comment upon this.

Line 134: At first I thought it odd to discuss a non-statistically significant result, but I do think it's a good point that there seem to be systematic differences between the groups when you look across the entire dose-response curve. At least for the proton leak and maybe even basal OCR – I'm not totally sure about the spare respiratory capacity. Perhaps there is a way to perform statistical comparisons between the curves. Or perhaps you could perform curve fits of Hill functions and see if there are statistically significant differences in the fit parameters (similar to what you did with EC50 comparisons in a later section). It may be worth consulting with a statistician to see if this would be okay, or if it would be considered p-hacking.

As a general comment, the SI seems like it could use a little more polish. If I wanted to reproduce your cybrid generation method, I would look to the SI to understand how to evaluate them. But unfortunately, the SI doesn't do a super great job of citing references that explain previously-used methods in better detail.

Line 535: text should be moved above the figure.

Line 544 Table: Looks like a formatting issue pushed some blank table space to the next page. Also, what is "custom"?

Line 553 Figure: It looks like ρ_0 is mis-labeled on the left figure.

Line 559: Section S4: please include a citation so readers could learn more about these methods.

[Editors’ note: further revisions were suggested prior to acceptance, as described below.]

Thank you for resubmitting your work entitled "The generation of HepG2 transmitochondrial cybrids to reveal the role of mitochondrial genotype in idiosyncratic drug-induced liver injury: a translational in vitro study" for further consideration by *eLife*. Your revised article has been evaluated by Mone Zaidi (Senior Editor) and a Reviewing Editor.

The manuscript has been improved but there are some remaining issues that need to be addressed, as outlined below:

Please address Comments from Reviewer #3. The limitations should be included in the Discussion section.

*Reviewer #3 (Recommendations for the authors):*

1) In this report, the authors describe the creation of transmitochondrial HepG2 cell cybrids to investigate the influence of variation in mitochondrial DNA on the susceptibility of cells to drug-induced toxicity. They propose that the production of such cell models has the potential to be representative of mitochondrial genome diversity across the population and their findings provide evidence that these mechanisms could play a role in individual susceptibility to hepatic adverse drug reactions. Although the results from their work only show modest differences between the haplogroups tested, this work is valuable to the field as it provides clear evidence that mtDNA variation influences susceptibility to drug-induced mitochondrial dysfunction and cell death. For many years the link between mtDNA and adverse drug reactions has been postulated by leaders in the field, and I believe that the publication of this work will enable rapid expansion in this area of research. Furthermore, the repurposing of transmitochondrial cybrid cells for studying drug safety brings a powerful new tool to toxicological research which could find utility in academia, and as the authors suggest, drug development. Overall, the data is robust and well analysed and the interpretations and conclusions made by the authors are appropriate.

2) In this work, the authors have generated several novel cell lines, each with distinct mitochondrial DNA content, to investigate whether individual variation in mitochondrial genotype can influence susceptibility to hepatotoxicity. The novelty of this model is its creation from HepG2 cells making it not only practical and reproducible but also relevant to preclinical testing during drug development. The results from this work indicate that variation in the mitochondrial genome does play a role in susceptibility to some hepatotoxins, but that this is a complex relationship that may only be one small factor in predisposing an individual to risk of an adverse drug reaction. In this report, only two classes of compounds are examined in the transmitochondrial cybrids, and the results from these suggest that these mechanisms are highly compound-specific. Therefore, it is difficult to extrapolate whether the same mechanisms would be evident across a wider variety of compounds. Another major limitation comes from the lack of diversity of mitochondrial haplogroups that are represented in the cybrids, which are made up of only haplogroup J and H. These two haplogroups are close phylogenetic relations. The addition of more diversity in the panel to increase phylogenetic coverage may uncover greater differences. Finally, the findings would be more robust, if transmitochondrial cybrids were created from different individuals, but of the same haplogroup subclade to see if the same effects would be replicated.

Overall summary - for many years the link between mitochondrial DNA and adverse drug reactions has been postulated by leaders in the field, and despite caveats mentioned above, this work provides a first proof of principle that this is a contributing factor and provides novel methodology and tools to enable this area of research to expand. Specifically, the repurposing of transmitochondrial cybrid cells for studying drug safety brings a powerful new tool to toxicological research which could find utility in academia, and as the authors suggest, drug development.

Specific feedback to the authors.

There are several points that the authors should consider:

1) The process of creating the HepG2 rho(0) cells using ethidium bromide has the potential to not only deplete mitochondrial DNA, but also damage genomic DNA. Have the authors any data to support the fidelity of the original HepG2 cell line? Furthermore, does the whole process (rho (0) generation and repopulation with mtDNA) have the potential to introduce differences between the cell lines so they no longer maintain the consistent HepG2 parental cell background that is essential for this model to be useful?

2) As a tumour-derived cell lineHepG2 cells suffer from a lack of physiological relevance. They contain low levels of both phase I and II drug-metabolising enzymes, transporters, and secondary structures. This limits the translatability of any findings to humans and also makes them unsuitable for testing any compounds where a metabolite of the drug is required for toxicity. Why did the authors not produce these models in a more physiologically relevant system e.g. HepaRG cells.

3) Are the test concentrations used in the manuscript relevant for the in vivo concentrations in man? The authors should provide these Cmax concentrations in the manuscript. This is important if this model is to be useful during preclinical development.

---

## [Author Response]

[Editors’ note: The authors appealed the original decision. What follows is the authors’ response to the first round of review.]

Comments to the Authors:In this manuscript, Amy Ball et al., use HepG2 cell-derived mitochondrial Cybrids to study the effects of drug-induced liver injury. The aim here is to demonstrate the role of varying mitochondrial DNA levels as an important factor in drug-induced cell injury. The authors use Cybrid cell lines derived from Rho zero HepG2 Cells and H- and J haplotypes of blood cells from human donors. The use of cybrid cell lines to test the role of mtDNA variations, both mutational load, and DNA level, is an excellent idea. However, this manuscript contains many serious defects in terms of concept and experimental approach.Weaknesses: A major weakness is the use of HepG2 cells which is derived from a human hepatic tumor. Its genotype and metabolic profile are substantially altered from the normal hepatic cells. As the authors noted, one example is the absence of many inducible microsomal CYPs. The effects of Hydroxyflutamide, the metabolic product on mitochondrial function are not the same as the effects of flutamide hydroxylated in cells by the action of endogenously expressed CYPs.

The limitations of HepG2 cells are commented upon in the manuscript and are widely acknowledged across the toxicology sector. However, despite these limitations, HepG2 cells remain an important mainstay of pharmaceutical preclinical testing for hepatotoxicity as long as the parameters of their “fit-for-purpose” are clearly defined. Therefore, their use in this study is appropriate. Moreover, due to the widespread use of HepG2 cells in preclinical toxicity testing, their conversion into cybrid cells is both highly relevant and useful to both the pharmaceutical industry and academic research, see our previous publication (Nature Reviews in Drug Discovery, 2020, PMID: 31748707). This statement has been included in the manuscript (pg 4, lines 17 – 20). Additionally, cybrid cells cannot be generated in the “gold standard” model for drug-induced liver injury, primary human hepatocytes, as these do not undergo division and rapidly de-differentiate over a 24 h period, see our previous publication (Arch Toxicol, 2017, PMID: 27039104).

Concerning whether it is appropriate to use 2-hydroxyflutamide to recapitulate the in vivo effects of hydroxyflutamide, it should be noted that the 2-hydroxy flutamide used in this study is structurally identical to the endogenous, CYP-mediated 2-hydroxylated metabolite. Specifically, upon administration, flutamide undergoes extensive first-pass metabolism, primarily by conversion to 2-hydroxyflutamide via cytochrome P450 1A2 (CYP1A2), followed by glucuronidation before excretion. It is known that following a single 250 mg dose of flutamide, its maximum plasma concentration (Cmax) is 72.2 nM, yet the Cmax of 2-hydroxyflutamide is 4.4 μM (PMID: 3169114). Flutamide is considered to be a pro-drug for 2-hydroxyflutamide. Therefore, given the identical chemical structures, and lack of any contrary literature upon this point, we disagree with this comment that “The effects of Hydroxyflutamide, the metabolic product on mitochondrial function are not the same as the effects of flutamide hydroxylated in cells by the action of endogenously expressed CYPs.” We have previously published using this in vitro model (Tox Sci, 2016, PMID: 27413113). It should be noted that the lack of CYP enzymes in this model allows us to specifically separate toxic mechanisms due to the parent compound and the major hydroxy metabolite. This point has been included in the revised manuscript (pg 5, lines 2 – 8).

Another important drawback is that the analysis is strictly dependent on respiratory flux analysis. It is not clear why mtDNA contents are not analyzed. To that matter it is not clear if the mtDNA contents in donor haplotypes H and J are indeed different and how much different. Using two haplogroups with unknown mtDNA contents is not sufficient. One needs to generate cell lines containing multiple levels of mtDNA under the same nuclear DNA background.

We would like to provide further clarification on this point. We have supposed that by “mtDNA contents”, the reviewer is referring to a need to quantify absolutely the mtDNA copy number within each cybrid cell line, and not the whole mitochondrial genome sequence which we do have for each of the cybrid cell lines created. We believe that this line of enquiry, looking at mtDNA content, is out of scope for the research presented herein, which aims to evaluate whether specific single nucleotide polymorphisms, and/or haplogroups, may play a contributory role in determining individual susceptibility to mitotoxins. That said, in our most recent projects utilising the cybrid cells we have quantified mtDNA content, and no significant differences in mtDNA levels were apparent at a basal level. Additionally, it is our experience that mtDNA content (in terms of copy number) is highly temporally variable. Finally, with regards to the additional suggestion to manipulate mtDNA levels in the cybrid cells, to further manipulate the mtDNA levels in these cybrid cells following their already extensive modification would not add to the scientific understanding of how specific single nucleotide polymorphisms, and/or haplogroups, contribute to individual susceptibility to mitotoxins.

In addition, we would like to point out that H and J are each a dataset which is comprised of distinct cybrid cell lines (5 haplogroup J cybrid lines and 5 haplogroup H cybrid lines) not a single cell line from each, which may have been assumed by the reviewer. Our newly created figures (refer to point 2.2) more clearly show the results from each cell line, rather than as a single average (mean) value. We believe that this represents the data and its variability more clearly.

Reviewer #1 (Recommendations for the authors):Line 87: Is haplogroup J more strongly linked to iDILI than haplogroup H? If so, please note and comment upon this.

Haplogroup J is reported to be more susceptible to toxic mechanisms for example see PMID: 29486301, however, no previous work has looked specifically at haplogroup J and idiosyncratic DILI. We have expanded on this point in the revised manuscript (pg 4, lines 24-28).

Line 134: At first I thought it odd to discuss a non-statistically significant result, but I do think it's a good point that there seem to be systematic differences between the groups when you look across the entire dose-response curve. At least for the proton leak and maybe even basal OCR – I'm not totally sure about the spare respiratory capacity. Perhaps there is a way to perform statistical comparisons between the curves. Or perhaps you could perform curve fits of Hill functions and see if there are statistically significant differences in the fit parameters (similar to what you did with EC50 comparisons in a later section). It may be worth consulting with a statistician to see if this would be okay, or if it would be considered p-hacking.As a general comment, the SI seems like it could use a little more polish. If I wanted to reproduce your cybrid generation method, I would look to the SI to understand how to evaluate them. But unfortunately, the SI doesn't do a super great job of citing references that explain previously-used methods in better detail.

Thank you for these comments, we have consulted with Mark Lennon, Director of Statistics at GSK, to review our analytical methods. He advised us that our previous analysis was not optimal due to its dependence on multiple comparisons, which reduced power. Therefore, we have reanalysed all data sets using a linear mixed model with the subject fitted as a random effect and the haplogroup fitted as a fixed effect. The dose-response relationship within each haplogroup was modelled using a polynomial to degree 3 or degree 2, as appropriate, and fitted to the log10 concentration. Post-hoc tests were then obtained from the fitted models to compare the mean response between the two haplogroups. For the models with the fitted polynomials, this comparison was made at the mean concentration i.e. in the middle of the fitted dose-response curve. See updated methodology (pg 33, lines 1 – 10).

Not only has this alternative analysis revealed additional relationships, but the figures now graphically represent the variation within each haplogroup as each cybrid cell is now plotted as a separate point on the graphs.

We believe that this further analysis has strengthened the interpretation and conclusions of the dataset and improved the manuscript.

Line 535: text should be moved above the figure.Line 544 Table: Looks like a formatting issue pushed some blank table space to the next page. Also, what is "custom"?Line 553 Figure: It looks like ρ_0 is mis-labeled on the left figure.Line 559: Section S4: please include a citation so readers could learn more about these methods.

We have corrected these errors. Please note that all supplementary material is now included in the main article. The generation and characterisation of the transmitochondrial cybrids, previously included as supplementary information, can now be found in the methods section.

[Editors’ note: what follows is the authors’ response to the second round of review.]

The manuscript has been improved but there are some remaining issues that need to be addressed, as outlined below:Please address Comments from Reviewer #3. The limitations should be included in the Discussion section.Reviewer #3 (Recommendations for the authors):Specific feedback to the authors.There are several points that the authors should consider:1) The process of creating the HepG2 rho(0) cells using ethidium bromide has the potential to not only deplete mitochondrial DNA, but also damage genomic DNA. Have the authors any data to support the fidelity of the original HepG2 cell line? Furthermore, does the whole process (rho (0) generation and repopulation with mtDNA) have the potential to introduce differences between the cell lines so they no longer maintain the consistent HepG2 parental cell background that is essential for this model to be useful?

This is an important consideration of these transmitochondrial cybrid cells and have included reference to this limitation in the discussion (21, lines 27 – 32). Although we have no data to support fidelity of HepG2 cells after EtBr treatment we describe the additional mitigation steps taken by us to minimise off-target cellular effects when generating the cybrid cells.

2) As a tumour-derived cell lineHepG2 cells suffer from a lack of physiological relevance. They contain low levels of both phase I and II drug-metabolising enzymes, transporters, and secondary structures. This limits the translatability of any findings to humans and also makes them unsuitable for testing any compounds where a metabolite of the drug is required for toxicity. Why did the authors not produce these models in a more physiologically relevant system e.g. HepaRG cells.

We are in agreement with the reviewer on this issue and note it ourselves (pg 22, line 19). However, the creation of cybrid cells from cell lines such as HepaRG is problematic due to the length of the process to create the cells, which is not compatible with HepaRG growth and differentiation protocols which must be strictly adhered to. For example, specifically, the cloning of HepaRG cells is not possible since any deviation from the recommended cell densities can induce the transformation of cells. We also think that there are some advantages of using HepG2 cells, and we have further described these in the discussion (pg 22, lines 23-25).

3) Are the test concentrations used in the manuscript relevant for the in vivo concentrations in man? The authors should provide these Cmax concentrations in the manuscript. This is important if this model is to be useful during preclinical development.

We thank reviewer 3 for this helpful comment. This information has now been added to the manuscript (page 34, lines 10 -13).